# Hydrogen bond based smart polymer for highly selective and tunable capture of multiply phosphorylated peptides

Guangyan Qing[1], Qi Lu[1], Xiuling Li[2], Jing Liu[2], Mingliang Ye[2], Xinmiao Liang[2] & Taolei Sun[1,3]

Multisite phosphorylation is an important and common mechanism for finely regulating protein functions and subsequent cellular responses. However, this study is largely restricted by the difficulty to capture low-abundance multiply phosphorylated peptides (MPPs) from complex biosamples owing to the limitation of enrichment materials and their interactions with phosphates. Here we show that smart polymer can serve as an ideal platform to resolve this challenge. Driven by specific but tunable hydrogen bonding interactions, the smart polymer displays differential complexation with MPPs, singly phosphorylated and non-modified peptides. Importantly, MPP binding can be modulated conveniently and precisely by solution conditions, resulting in highly controllable MPP adsorption on material surface. This facilitates excellent performance in MPP enrichment and separation from model proteins and real biosamples. High enrichment selectivity and coverage, extraordinary adsorption capacities and recovery towards MPPs, as well as high discovery rates of unique phosphorylation sites, suggest its great potential in phosphoproteomics studies.

[1] State Key Laboratory of Advanced Technology for Materials Synthesis and Processing, Wuhan University of Technology, 122 Luoshi Road, Wuhan 430070, China. [2] Key Laboratory of Separation Science for Analytical Chemistry, Dalian Institute of Chemical Physics, Chinese Academy of Sciences, 457 Zhongshan Road, Dalian 116023, China. [3] School of Chemistry, Chemical Engineering and Life Science, Wuhan University of Technology, 122 Luoshi Road, Wuhan 430070, China. Correspondence and requests for materials should be addressed to Xiuling Li (email: lixiuling@dicp.ac.cn) or to Xinmiao Liang (email: liangxm@dicp.ac.cn) or to T.S. (email: suntl@whut.edu.cn)

As one of the most common post-translational modification (PTM) types, reversible protein phosphorylation plays crucial regulatory roles in almost all aspects of cellular processes[1, 2]. Abnormal phosphorylation is closely associated with a number of human diseases, such as Alzheimer's disease (AD)[3] and cancers[4]. Large-scale phosphoproteome analysis suggests that the great majority of intracellular proteins simultaneously harbour more than one phosphate groups at various sites[5], and these multiple phosphorylation sites are not distributed randomly, but are often clustered in a specialized domain on a protein. Nearly 54% of all pSer/pThr sites are located within four amino acid residues of each other[6]. One of the most typical examples is microtubule-associated protein—Tau, abnormally hyperphosphorylated Tau, particularly the multisite phosphorylation near tubulin-binding domain, can accumulate into insoluble paired helical filaments, generating numerous neurofibrillary tangles that are regarded as one of the pathologic features of AD[7]. In addition to pathological significance, there are many examples where adjacent multisite phosphorylation regulates protein activities[8–10]. In the classical mitogen-activated protein kinase (MAPK) cascade, the activation of extracellular-signal-regulated protein kinases ERK1 and ERK2 is only triggered when they are phosphorylated at Thr183 and Tyr185 sequentially by MAPK kinase-1[9]. In a recent report[11], the transcription factor Elk-1 could be phosphorylated by the protein kinase ERK at eight transcriptional activation domain sites. Data showed that different rates at adjacent multisite phosphorylation sites of Elk-1 led to variety of protein activities. Phosphorylation of fast sites (Thr369 and Ser384) or intermediate sites (Thr354, Thr364, and Ser390) promoted mediator recruitment and transcriptional activation, while the modification of slow sites (Thr337, Thr418, and Ser423) counteracted both functions. Owing to great significance of adjacent multisite phosphorylation, it is highly desirable to capture and analyse the core MPPs[12].

Mass spectrometry (MS) is a mainstreamed analysis method in proteomics[13]. However, direct studies of MPPs using MS are difficult owing to their very low abundance in lysates of biosamples, significant background interference deriving from abundant non-modified peptides (NMPs) and singly phosphorylated peptides (SPPs), and low fragmentation efficiency in collision-induced dissociation[14]. Therefore, efficient enrichment and separation of MPPs from complex biosamples is the prerequisite to study their functions and relevant biochemical processes[15–17]. Conventional methods to capture phosphorylated peptides (PPs), such as metal oxide (e.g., $TiO_2$ or $ZrO_2$) affinity chromatography (MOAC)[18] and immobilized metal ion (e.g., $Fe^{3+}$ or $Ti^{4+}$) affinity chromatography (IMAC)[19], are based on chelation and electrostatic interactions between metal oxides or metal ions and negatively charged phosphates in PPs[20]. SPPs can be captured and identified efficiently using these methods[21]. Nevertheless, multiple chelation interactions exponentially increase the binding affinity of MPPs with materials and such chelation interactions cannot be easily regulated, both of which make the bound MPPs difficult to be dissociated and eluted from the material surfaces. In addition to the elution problem, IMAC materials also suffer from serious nonspecific adsorption to acidic interference peptides, which are highly abundant, and thus, the selectivity of these materials in practical applications is limited[22]. As a result, huge amounts of multiple phosphorylation sites are missed since only a small proportion of MPPs can be efficiently captured and then be detected by MS. Due to the lack of efficient enrichment tools, research on multisite phosphorylation lags far behind that of single-site phosphorylation. For example, in phosphorylation site database, such as PHOSIDA and UniProt, the great majority of identified sites are from SPP species.

Here we report a hydrogen bonding (H-bonding)-based smart copolymer[23, 24] approach to solve the problem of MPP enrichment and separation. The copolymer contains H-bonding-based phosphate recognition units, being incorporated into a flexible poly(N-isopropyl-acrylamide) (PNIPAAm) network. For the design of phosphate recognition unit, H-bonding interaction may display unique features compared with chelation interaction. Although an individual H-bond is very weak, the binding can be strong and highly specific when multiple complementary H-bonds are involved[25, 26]. This results in excellent tunability and controllability of the interactions over a wide dynamic range via rational molecular design. PNIPAAm is a well-known thermoresponsive polymer with a smart H-bonding network[27]. Its polymer chains can undergo reversible coil-to-globule conformational transitions upon external thermostimulus, leading to dramatic switching in macroscopic properties (i.e., morphology and wettability) of materials[28]. By incorporating appropriate biomolecule-recognition units into PNIPAAm, the polymer conformation may be intelligently modulated using guest biomolecules, dramatic conformational transition of the polymer chains will in turn remarkably influence the binding or release of the guest biomolecules[29–31]. Such copolymer may provide an efficient tool to modulate the dynamic adsorption/desorption behaviours of MPPs on material surfaces, which suggests an ideal platform for MPP enrichment and separation.

## Results

**Copolymer design.** For this purpose, we designed an H-bonding-based phosphate receptor, 4-(3-acryloyl-thioureido)-benzoic acid (ATBA; Supplementary Figs. 1, 2), to copolymerize with-NIPAAm to form a linear random copolymer (denoted as PNI-co-ATBA; Fig. 1b and Supplementary Fig. 3) through atom transfer radical polymerization[32] (ATRP; the detailed preparation method is described in Supplementary Methods). First, hydrogen, carbon, and hydrogen-carbon correlation nuclear magnetic resonance[33] ($^1H$, $^{13}C$, and $^1H$-$^{13}C$ COSY NMR) spectroscopy (Fig. 1d and Supplementary Figs. 4–6), fluorescence titration experiments[34] (Supplementary Figs. 7–9 and Supplementary Tables 1–3) and theoretical calculation (Fig. 1c and Supplementary Fig. 10) indicated that the ATBA functional monomer effectively bound hydrogen phosphates via H-bonds from thiourea and carboxyl moieties. The binding was highly selective, such that the association constant ($K_a$) of fluorescein-labelled thioureido benzoic acid (Supplementary Fig. 11) with hydrogen phosphate ($K_a$: $5.34 \times 10^4$ L mol$^{-1}$) in $H_2O$ was seven times higher than that with acetate (main source of acidic peptide, $K_a$: $7.6 \times 10^3$ L mol$^{-1}$) in Tris-HCl buffer solution. The $K_{a(\text{hydrogen phosphate})}/K_{a\ (\text{acetate})}$ ratio remarkably increased to 45:1 when the binding affinity of these anions with ATBA was evaluated in an acetonitrile ($CH_3CN$)/$H_2O$ (v v$^{-1}$ = 80:20) Tris-HCl buffer solution (corresponding to the real PP adsorption or enrichment condition, Supplementary Fig. 8 and Supplementary Table 2), which revealed the satisfactory discrimination capacity of the ATBA receptor towards hydrogen phosphate and acetate. In addition, the terminal carboxyl group in ATBA afforded substantial tunability to the binding affinity towards various PPs when the pH value of the solution was adjusted (Fig. 1e, Supplementary Figs. 12–14, and Supplementary Table 4).

When ATBA recognition units were incorporated into the copolymer, the flexible copolymer chains would provide multiple binding sites to combine with SPPs and MPPs differentially. This idea was proven by surface plasmon resonance adsorption experiment, in which the model NMP, SPP, and MPP displayed distinct association rate constants adsorbed on the copolymer surfaces (Supplementary Fig. 15). Moreover, the different

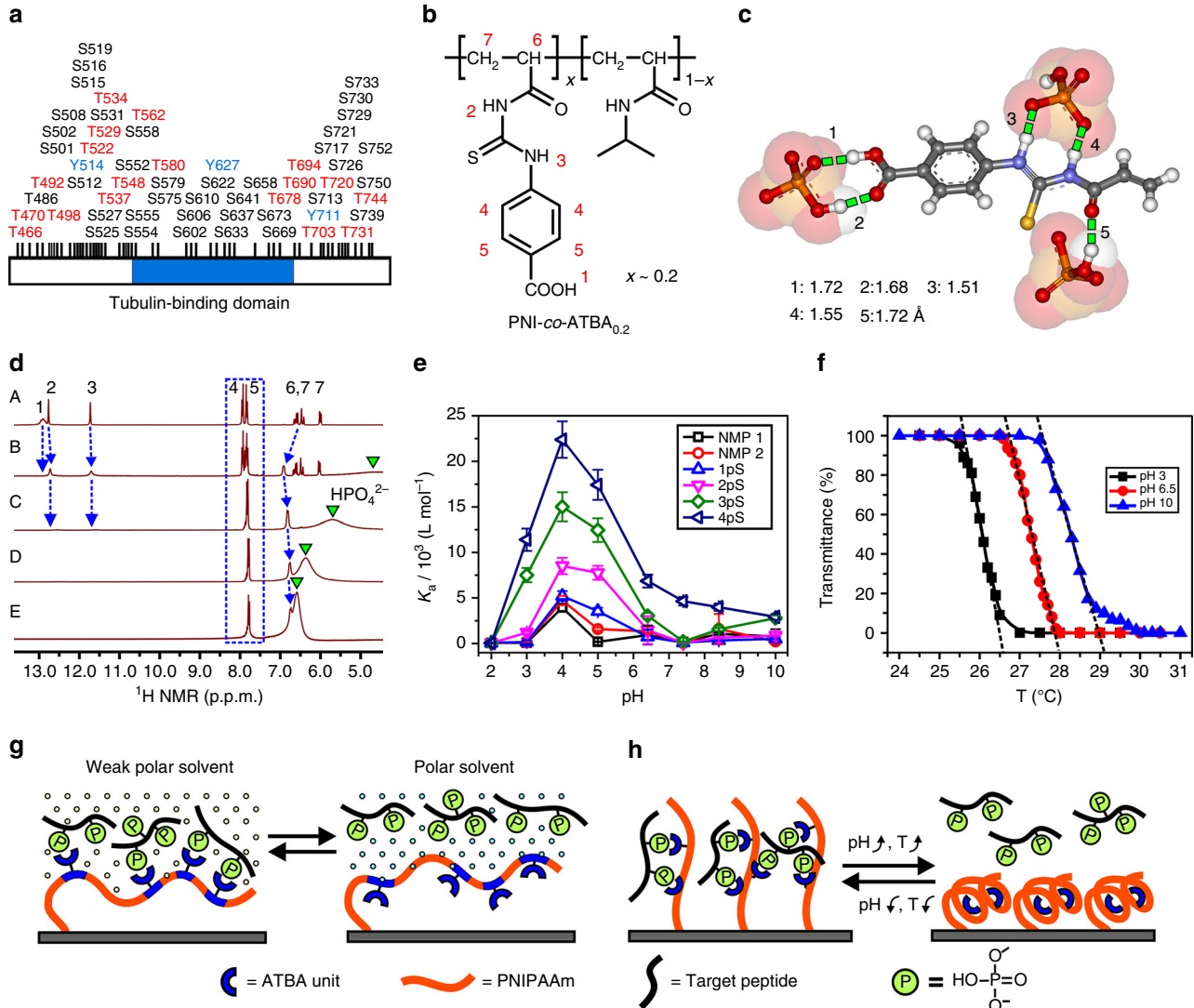

**Fig. 1** Design of smart copolymer. **a** Abundant phosphorylation sites located near the tubulin-binding domain of a microtubule-associated protein—Tau; the abnormal hyperphosphorylation of Tau is closely associated with several neurodegenerative disorders. Panel **a** is drawn based on the site information collected in PhosphoSitePlus database. **b** Chemical structure of poly(*N*-isopropyl-acrylamide-*co*-ATBA, denoted as PNI-*co*-ATBA. **c** Possible binding model of an ATBA functional monomer with three hydrogen phosphates ($HPO_4^{2-}$) driven by H-bonding interactions, as indicated by *green dashed lines* with different bond lengths. This model was obtained from quantum chemistry calculation (Gaussian, density functional theory (DFT), at 6-311 G level of theory, solvent: $H_2O$). **d** Hydrogen nuclear magnetic resonance ($^1H$ NMR) spectra of ATBA (A) upon additions of 0.5 (B), 1 (C), 2 (D) and 3 (E) equimolar amounts of $HPO_4^{2-}$ (tetrabutyl-ammonium as countercation) in deuterated dimethylsulphoxide at 20 °C. The chemical shift changes of ATBA protons (the attribution of each proton is shown in **b**) and $HPO_4^{2-}$ are indicated by *blue dashed arrows/box* or *green inverted triangles*, respectively. **e** pH-dependent association constant ($K_a$) of ATBA bound to NMPs 1 and 2 and four serine PPs with mono-, di-, tri-, or tetra-phosphates (i.e., 1pS–4pS), the peptide sequences are shown in Fig. 2l. $K_a$ values were obtained from fluorescence titration experiments performed in various buffer solutions and these peptides were labelled with fluorescein at N terminus. **f** Temperature-dependent transmittance change of the copolymer aqueous solution at pH 3, 6.5 or 10, which indicated the good thermoresponsiveness and pH responsiveness of the copolymer. **g** Schematic of reversible overturn of copolymer chains and the corresponding binding/release towards MPPs modulated by solvent polarity. **h** Schematic of reversible coil-to-globule transition of copolymer chains and the binding/release of MPPs modulated by solution pH or temperature

compatibilities of the poly(ATBA) and PNIPAAm segments with strong polar (i.e., $H_2O$) or weak polar solvents (i.e., $CH_3CN$) would make it possible to modulate the motion and overturn[35] of the copolymer chains by altering the solvent polarity (Supplementary Fig. 16). In addition, pH and temperature changes would also induce coil-to-globule conformational transitions of the copolymer[31, 36] (Fig. 1f and Supplementary Fig. 17). Therefore, this molecular design offered large flexibility in controlling the behaviours of copolymer chains by modulating the environmental conditions, which would further determine differential binding behaviours towards MPPs and SPPs. By this way,

high-efficiency enrichment of MPPs and separation of them from SPPs and NMPs could be realized via a chromatographic protocol with a stepwise elution procedure (Fig. 1g, h).

**Tunable adsorption behaviours of MPPs on smart copolymer surfaces.** The dynamic adsorption behaviours of MPPs on the PNI-*co*-ATBA$_{0.2}$ surfaces were monitored using a quartz crystal microbalance with dissipation (QCM-D), which can provide real-time information about variation in the mass and viscoelastic properties of the polymer film[37, 38]. Figure 2a–i display the

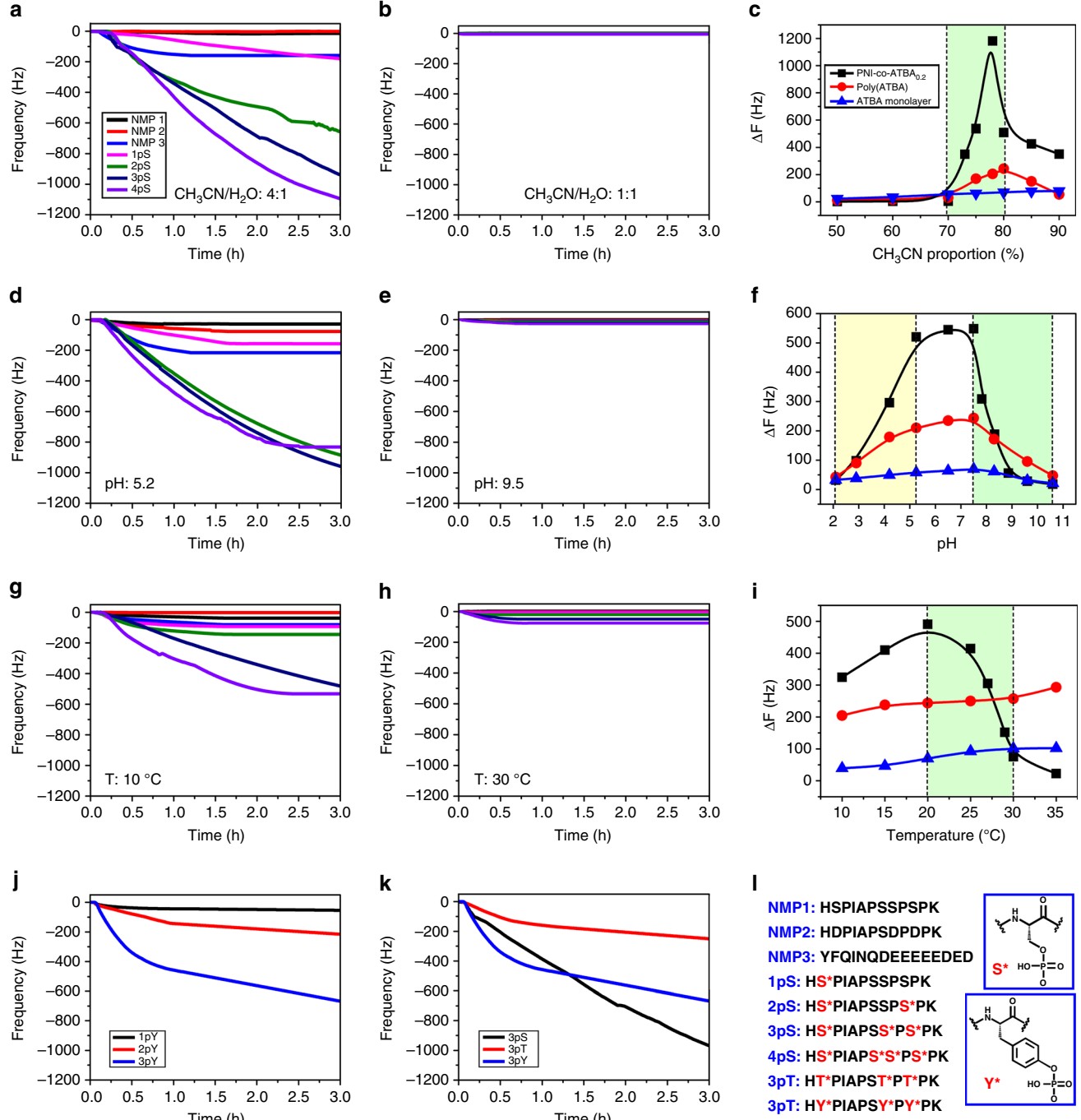

**Fig. 2** Selective and tunable adsorption of MPPs on copolymer film. **a**, **b**, **d**, **e**, **g**, **h** Dynamic adsorption curves of various PPs (1pS–4pS) and NMPs adsorbed on the PNI-*co*-ATBA$_{0.2}$ film, measured by QCM. **a**, **b** Peptide solutions with consistent pH of 7.4 and temperature of 20 °C, but different solvent polarities: (**a**) CH$_3$CN/H$_2$O v v$^{-1}$ = 4:1 or (**b**) CH$_3$CN/H$_2$O v v$^{-1}$ = 1:1. **d**, **e** Consistent solvent polarity of CH$_3$CN/H$_2$O v v$^{-1}$ = 4:1 and temperature of 20 °C, but different solution pH conditions: (**d**) pH: 5.2 or (**e**) pH: 9.5. **g**, **h** Consistent solvent polarity of CH$_3$CN/H$_2$O v v$^{-1}$ = 4:1 and pH of 7.4, but different temperatures: (**g**) 10 °C or (**h**) 30 °C. **c**, **f**, **i** CH$_3$CN proportion- (**c**), pH- (**f**) or temperature (**i**)-dependent QCM resonator frequency change (Δ*F*) in response to the adsorption of 4pS on PNI-*co*-ATBA$_{0.2}$ (*black*), poly(ATBA) (*red*) or ATBA monolayer (*blue*) surface, respectively. **j** Dynamic adsorption curves of tyrosine-modified PPs with mono-, di- or tri-phosphates (1pY–3pY) adsorbed on the PNI-*co*-ATBA$_{0.2}$ film in CH$_3$CN/H$_2$O (v v$^{-1}$ = 4:1) at pH 7.4 and 20 °C. **k** Adsorption curves of serine- (*black*), threonine- (*red*) or tyrosine- (*blue*) modified tri-PPs (3pS, 3pT and 3pY) adsorbed on the PNI-*co*-ATBA$_{0.2}$ film in CH$_3$CN/H$_2$O (v v$^{-1}$ = 4:1) at pH 7.4 and 20 °C. **l** Peptide sequences of NMPs and various PPs tested in this study. The phosphate-modified amino acids are indicated by *red characters* and their corresponding chemical structures

dynamic adsorption curves of model NMPs (NMP1–3), serine mono-, di-, tri-, and tetra-PPs (abbreviated as 1pS–4pS) on QCM crystals grafted with our copolymer (ATBA content: ~ 20%; film thickness: 26 ± 1 nm). These model peptides have identical amino acid sequences but only differ in the numbers of phosphate

groups, as illustrated in Fig. 2l. NMP2 is a specially designed acidic peptide to mimic the main acidic interference peptides in PP enrichment, in which Ser2, Ser8, and Ser10 residues in NMP1 were substituted by Asp. Meanwhile, in real biosamples, acidic peptides with multiple Asp and/or Glu residues right next to each

other are abundant and will strongly interfere the MPP enrichment. To evaluate the anti-interference capacity of our material towards these acidic peptides, three special NMPs were prepared, the most typical one is listed as NMP3 with nine adjacent Asp or Glu residues. Then, different solvent polarity, solution pH, and temperature were used in order to modulate the adsorption behaviours of MPPs on the copolymer surfaces.

**Differential adsorption behaviours of MPPs, SPPs and NMPs.** First, the effect of solvent polarity on MPP adsorption was discussed. $CH_3CN/H_2O$ mixture is a common mobile phase in liquid chromatography and solvent polarity can be conveniently adjusted by changing the proportion of $CH_3CN/H_2O$. At a $CH_3CN$ content of 80% (Fig. 2a) or higher, the SPP and MPPs (i.e., 1pS–4pS) exhibited strong but differential adsorption to the copolymer film with maximum frequency change ($\Delta F$) varying from ~300 Hz to more than 1200 Hz. Under the same condition, for both NMP1 and NMP2, the QCM adsorption was almost negligible ($\Delta F < 10$ Hz). When the acidic NMP3 and other two specially designed NMPs were evaluated, the maximum $\Delta F$ values were still lower than 200 Hz (Supplementary Fig. 18), which indicated satisfactory discrimination capacity of our material towards MPPs, SPPs and NMPs.

As a comparison, the adsorption performance of several common PP enrichment materials was evaluated. For example, the QCM crystals were covered with a thin layer of $TiO_2$ or $ZrO_2$ (purchased from Q-Sense Corp., Sweden) in order to mimic the MOAC materials. However, on $TiO_2$ or $ZrO_2$ surfaces, the 1pS–4pS adsorption-induced frequency changes ($\Delta F$ values) were all < 25 Hz under the optimized conditions[39], and the difference among the SPP and MPPs was subtle (Supplementary Figs. 19–22). Furthermore, commercially available $TiO_2$ microspheres (purchased from GL Sciences, Tokyo, Japan; Supplementary Fig. 23) or a 2,2′-azanediyl-diacetate-$Fe^{3+}$ complex (representing IMAC materials; Supplementary Fig. 24) were immobilized on the QCM crystals to perform the control experiment, the MPP adsorption-induced frequency changes was also low ($\Delta F_{max} < 170$ Hz). These results illustrated the obvious advantage of our copolymer in satisfactory selectivity and high adsorption capacities towards MPPs.

**Solvent polarity modulated MPP adsorption.** In addition, the copolymer displayed highly tunable adsorption towards MPPs via altering solvent polarity. When the $CH_3CN$ content was decreased to 70% (Fig. 2b) or lower, the adsorption of all PPs decreased sharply and $\Delta F$ values of <10 Hz were observed. We further used 4pS as an example to discuss the impact of solvent polarity on MPP adsorption in detail. Notably, a dramatic change in adsorption quantity was observed with $CH_3CN$ content changing from 70 to 80% (Fig. 2c and Supplementary Fig. 25). This narrow adsorption conversion window is fundamentally different from the typical hydrophilic interaction modes in liquid chromatography, which usually shows an approximately linear relationship between solvent polarity and chromatographic retention capacity of analyte[40]. Benefiting from this new feature, MPP adsorption and desorption process could be switched flexibly by solvent polarity.

**pH modulated MPP adsorption.** Solution pH was also found to influence MPP adsorption process. Isoelectric focusing electrophoresis indicated that the isoelectric point of the copolymer was ~5.8; thus, the ionization states of the ATBA units changed considerably at pH values below and above 5.8. Accordingly, alkaline solutions (pH 7.4–11) reduced the binding affinity of the copolymer towards MPPs because of electrostatic repulsion

between the negatively charged carboxyl groups of ATBA and the phosphates of MPPs. By contrast, weakly acidic solutions (pH 4–7) promoted the complexation due to the enhanced H-bonding interaction between the positively charged thiourea groups of ATBA and MPPs. When a strongly acidic solution (pH 2–3) was applied, the delicate copolymer H-bonding network would collapse, resulting in a sharp decrease in binding affinity with MPPs. Importantly, solution pH also affected the coil-to-globule transition of the copolymer chains[28, 29, 36]. Under weakly acidic conditions (pH 4–7), the copolymer chains maintained a relaxed and extended conformation, providing abundant binding sites for MPPs. Under basic conditions (pH 7.5–10), the contracted conformation of the copolymer chains prevented the sufficient contact and efficient binding between ATBA units and MPPs. This presumption was consistent with the QCM experiments carried out under different pH conditions, as shown in Fig. 2d, e. Scanning the adsorption behaviours of 4pS over a broad pH range revealed two distinct adsorption conversion windows for different adsorption modes (Fig. 2f and Supplementary Fig. 26). The first window (pH 2.0–3.0) corresponded to the decomposition of copolymer H-bonding network under strongly acidic conditions, whereas the second window (pH 7.5–8.8) was attributed to the dramatic conformational transition of copolymer chains induced by altering solution pH. These data indicated that the tightly bound MPPs could be dissociated from the copolymer surfaces easily by increasing or decreasing the solution pH value.

**Temperature modulated MPP adsorption.** As a typical thermosensitive copolymer, temperature strongly influenced the conformation transition of the copolymer chains and their adsorption capacities towards MPPs. Cloud-point assay using an ultraviolet spectrophotometer showed that the lower critical solution temperature of the $PNI$-$co$-$ATBA_{0.2}$ was ~28 °C in pure water (Fig. 1f and Supplementary Fig. 17). QCM adsorption experiment performed at different temperature demonstrated a remarkable change in the adsorption quantity of 4pS occurred between 20 and 30 °C (Fig. 2i and Supplementary Fig. 27). Below 28 °C, the copolymer chain exhibited a relaxed conformation and the adsorption of 4pS was strong (Fig. 2g), whereas above 30 °C, the copolymer chains were contracted and the adsorption of 4pS was extremely weak (Fig. 2h). The aforementioned analysis clearly revealed that solvent polarity, solution pH and temperature became three efficient parameters for modulating the adsorption behaviours of MPPs on the smart copolymer surfaces.

**Control experiment to validate the rationality of the copolymer design.** Poly(ATBA) homopolymer, ATBA monolayer and pure PNIPAAm were introduced to perform the MPP adsorption experiments under the same conditions. For poly(ATBA) film and ATBA monolayer, the adsorption of 4pS on these surfaces was not affected by solvent polarity or temperature, but was slightly affected by solution pH contributed by the carboxyl groups in ATBA molecules (*red and blue curves* in Fig. 2c, f, i). In addition, the PNIPAAm film did not display any adsorption to SPPs and MPPs, which indicated that the strong adsorption of $PNI$-$co$-$ATBA_{0.2}$ towards MPPs originated from chemical binding rather than physical adsorption[41]. Furthermore, control experiments using a series of copolymer films based on ATBA analogues (Supplementary Figs. 28 and 29) revealed that carboxyl, thiourea and phenyl groups in ATBA integrated into one system, and the absence of any component would lead to a sharp decrease in the adsorption capacity towards MPPs. In addition, 20% was determined to be the optimal ATBA content for the copolymer. For the lower ATBA contents (i.e., 10 or 15%), substantially

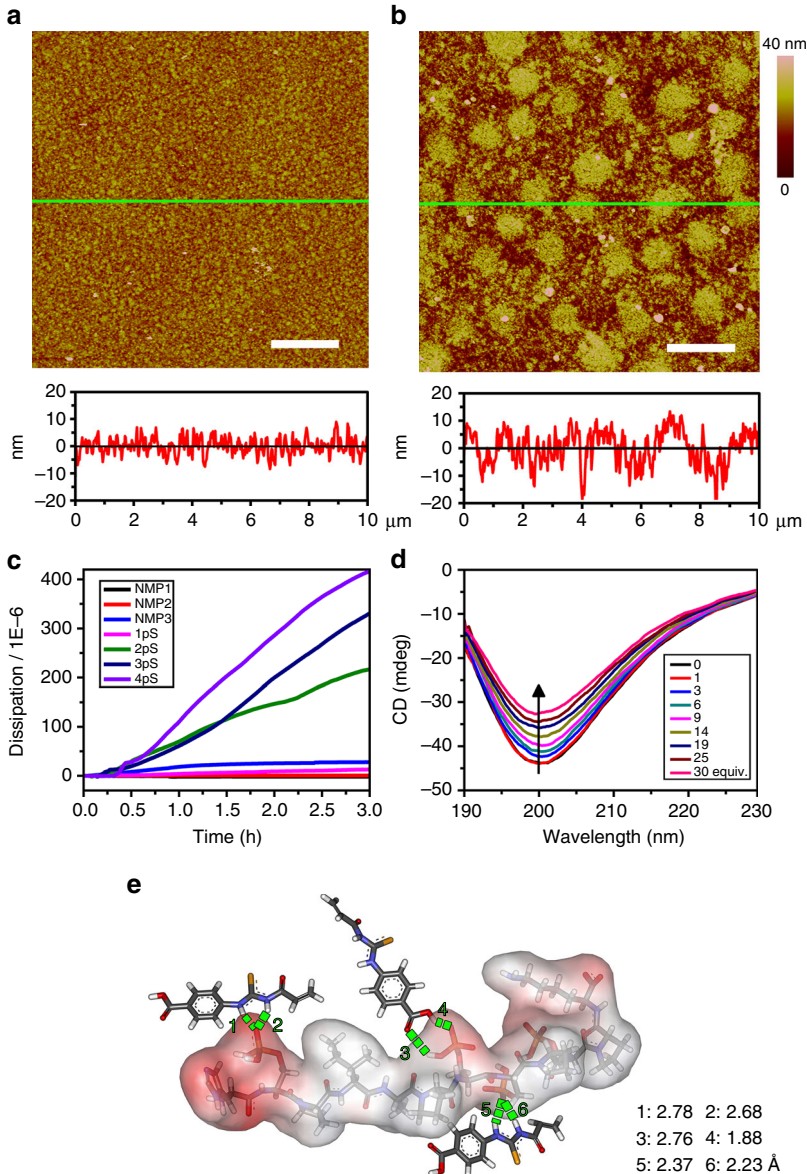

**Fig. 3** Mechanism analysis of the binding between copolymer and MPPs. **a**, **b** AFM images of PNI-*co*-ATBA$_{0.2}$ film before (**a**) and after (**b**) treatment with a 4pS solution (concentration: 37.5 μmol L$^{-1}$ in Tris buffer at pH 7.4 and 20 °C) as well as the corresponding section profiles along the *green lines*. *Scale bars*: 2 μm. **c** Time-dependent dissipation curves of the copolymer film in response to the adsorption of various PP (1pS–4pS) and NMP (1–3), solution: CH$_3$CN/ H$_2$O (v v$^{-1}$) = 4:1, pH = 7.4 at 20 °C. **d** CD spectra of 4pS upon additions of different (0–30) equimolar amounts of ATBA monomers in water. **e** Possible binding mode of 4pS with three ATBA monomers, obtained from quantum chemistry calculations, density function theory, at 6–31 G level of theory, solvent: water, temperature: 20 °C

weaker 4pS adsorption on the copolymer films were detected due to the insufficient PP binding groups in the copolymer chains (Supplementary Fig. 30). By contrast, higher ATBA contents (i.e., 25 or 35%) remarkably strengthen the polymer H-bond network, leading to a contracted polymeric film (Supplementary Fig. 31). Consequently, only a small amount of ATBA groups remain exposed and can bind to exterior PPs, resulting in a sharp decrease in the MPP adsorption performance of the copolymer film. Above results illustrated the rationality of the copolymer design, which allowed for dynamic manipulation of the MPP adsorption in a large extent. This excellent tunability may facilitate the optimization of MPP enrichment conditions.

**Selective adsorption to MPPs with phosphorylation occurred at different amino acids**. In addition to serine, which is the most

commonly phosphorylated amino acid, threonine and tyrosine are also often phosphorylated. In particular, tyrosine phosphorylation of specific enzymes regulates insulin and growth factor signalling to glucose transport, glycogen synthesis, protein synthesis and transcriptional activation[42, 43]. Interestingly, our copolymer film displayed strong but differential adsorption to tyrosine mono-, di- and tri-PPs (1pY–3pY; Supplementary Fig. 20 and Fig. 2j). Moreover, after 1 h, the capacity of our material to adsorb 3pY (8.11 μg cm$^{-2}$) was higher than its capacity to adsorb serine tri-PP (3pS, 6.89 μg cm$^{-2}$) or threonine tri-PP (3pT, 2.86 μg cm$^{-2}$) (Fig. 2k). Mechanism analysis (Supplementary Figs. 6, 7e and 32 and Supplementary Table 1) indicated that this difference mainly originated from different binding affinity of the ATBA molecule towards phenyl phosphate ($K_a$: 7.8 × 10$^4$ L mol$^{-1}$) and hydrogen phosphate ($K_a$: 5.34 × 10$^4$ L mol$^{-1}$). Distinct QCM adsorption curves (Fig. 2k and Supplementary Figs. 33 and 34)

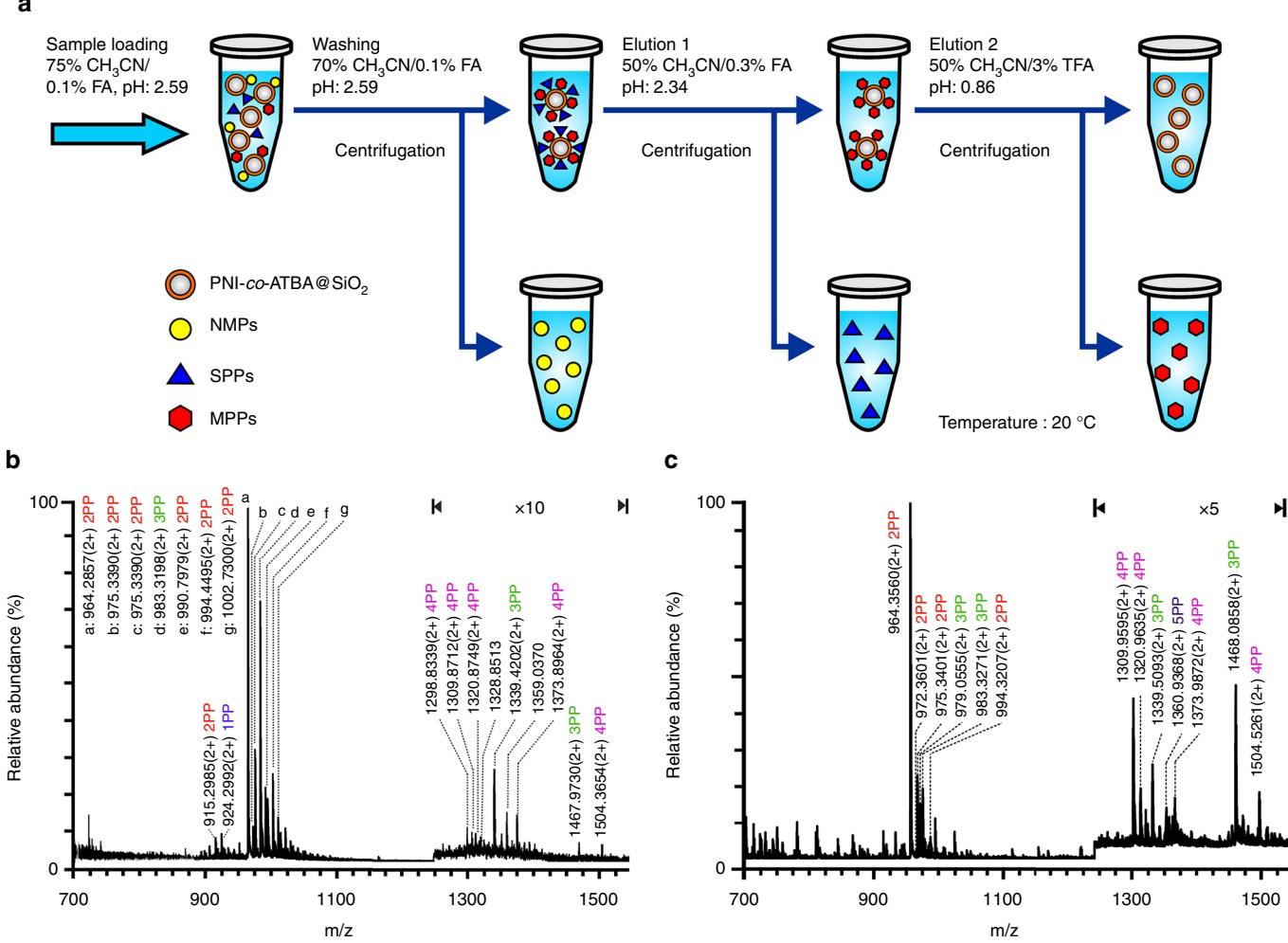

**Fig. 4** Enrichment and separation of MPPs in model protein samples. **a** Schematic illustration of PP enrichment strategy based on a dSPE mode. **b**, **c** MS of PPs enriched with PNI-*co*-ATBA$_{0.2}$@SiO$_2$ from tryptic digests of casein and BSA at molar ratios of 1:100 (**b**) and 1:500 (**c**). Peptides are labelled with their *m/z* values, mono-, di-, tri-, tetra-, penta-PPs are marked with *blue*, *red*, *green*, *purple* and *dark blue characters*, respectively. The *y*-axis in panel **b** or **c** represents relative abundance in MS. The base peak (the tallest peak) of a particular ion is normalized to 100%. The other peaks in MS appear between 0 and 100% abundance. Detailed information about peptide sequences and phosphorylation sites is shown in Supplementary Table 5

further verified that our material interacted differently with PPs depending on the amino acids where the phosphorylation occurred. This result illustrated another attractive feature of our copolymer that conventional MOAC or IMAC materials lack, which usually displayed unbiased adsorption[20] towards S, T and Y phosphorylation owing to the intrinsic characteristics of chelation interactions of these materials.

**MPP adsorption induced surface property transition of the copolymer film**. Surface properties of copolymer film were strongly influenced by PP adsorption. Corresponding changes in conformation, thickness, and viscoelasticity of the copolymer film were recorded simultaneously as dissipation curves by QCM-D[38]. As shown in Fig. 3c, distinct dissipation curves (reflected as typical chemical adsorption processes) were observed upon adsorption of diverse PPs. Substantial increases in dissipation were observed for the adsorption of MPPs (i.e., 2pS, 3pS and 4pS) compared to those of SPP (i.e., 1pS) or NMPs (i.e., NMP1 and NMP2). The maximum dissipation change ($\Delta D$) was induced by 4pS with a value of $4.16 \times 10^{-4}$. According to the classical QCM adsorption theory[38, 41], this remarkable dissipation increase suggested that the copolymer chains stretched into relaxed and swollen states after interacting with 4pS (Supplementary Fig. 35).

We next used atomic force microscopy (AFM)[44] to observe morphological change of the copolymer film before and after immersion in a 4pS solution (37.5 µmol L$^{-1}$ in Tris-HCl buffer, pH 7.4) for 1 h. As shown in Fig. 3a, b, upon interaction with 4pS, the copolymer film exhibited remarkable expansion, generating a large number of expansion bulges. Meanwhile, the copolymer film thickness substantially increased from ~19 ± 1 to 33 ± 2 nm. Under the same condition, the film thickness only increased to 24 ± 2 nm upon interaction with 1pS (Supplementary Figs. 36 and 37), which further confirmed dramatic conformational changes of the copolymer chains upon the adsorption of 4pS. Consistent morphological change was observed when the copolymer film was treated with a CH$_3$CN/H$_2$O (v v$^{-1}$ = 80:20) mixture containing 4pS (Supplementary Fig. 38). Meanwhile, similar changes in dissipation curves and copolymer film morphology were also observed for 3pT and 3pY (Supplementary Figs. 39 and 40).

**Conformational changes of MPPs after interacting with ATBA**. Conformational changes of 4pS upon interaction with ATBA were monitored using circular dichroism (CD) (Fig. 3d)[45]. In H$_2$O, 4pS displayed a strong negative CD spectral peak centred at 200 nm, attributing to the α-helix structure of 4pS. Upon the

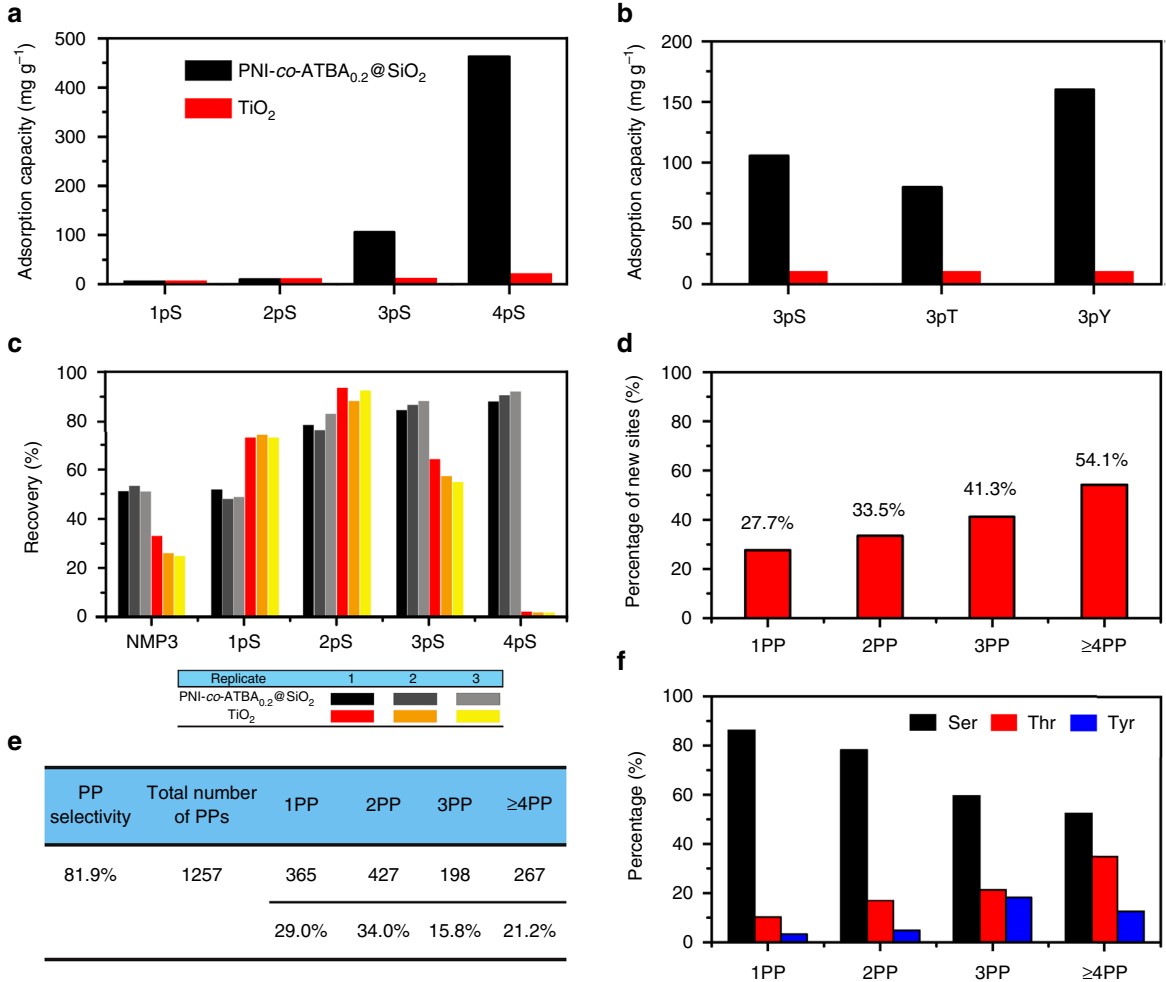

**Fig. 5** Adsorption capacities and identified unique phosphorylation site information. **a, b** Comparison of adsorption capacities of PNI-*co*-ATBA$_{0.2}$@SiO$_2$ (*black columns*) and commercially available TiO$_2$ (*red columns*) towards 1pS–4pS (**a**), or towards 3pS, 3pT and 3pY (**b**). **c** Comparison of recovery of PNI-*co*-ATBA$_{0.2}$@SiO$_2$- (*black and gray columns*) and TiO$_2$- (*red, orange and yellow columns*) based enrichment methods towards 1pS–4pS, obtained from three parallel MS measurements. **d, e** Percentages of unique phosphorylation sites (**d**) or distribution proportions (**e**) for mono-, di-, tri- or not less than tetra-PPs enriched by our material from tryptic digests of HeLa S3 cell lysates. **f** Comparison of relative ratios of serine, threonine to tyrosine phosphorylation for mono-, di-, tri- or not less than tetra-PPs, identified by the polymeric material from the same cell sample. For panels **d**, **e**, **f**, the number of peptides and phosphorylation sites for 1PP–4PP was calculated from the pool of three analytical repeats

addition of ATBA aqueous solution (CD silent), the intensity of the α-helix peak decreased gradually from −44 to −32 mdeg, which revealed that the 4pS conformation was altered slightly upon the complexation with ATBA. Similar conformational change was also revealed by bioattenuated total reflectance Fourier transform infrared (IR) spectroscopy in deuterated water (Supplementary Figs. 41 and 42). In addition, quantum chemistry calculation[46] described a possible binding model of 4pS with three ATBA molecules (Fig. 3e and Supplementary Fig. 43) in H$_2$O at 20 °C. In this model, ATBA precisely recognized the phosphate domains in 4pS, and six sets of intensive H-bonding interactions were formed between ATBA and 4pS. Similarly, the binding models of 3pT or 3pY with ATBA molecules were also investigated (Supplementary Figs. 44–47).

**MPP enrichment from model protein samples.** Above analysis revealed the good potential of our copolymer in capture of MPPs. To validate this point, PNI-*co*-ATBA was grafted onto the surface of porous silica microspheres (particle size: 5 μm, inner pore size: 300 Å) through surface-initiated ATRP[32],

used as a chromatographic stationary phase (denoted as PNI-*co*-ATBA$_{0.2}$@SiO$_2$). The characterization data of this material is shown in Supplementary Fig. 48, the grafting ratio of copolymer was 8% calculated from thermogravimetric analysis and ATBA content in the copolymer was ~20%. Using tryptic digests of α-casein as a standard phosphoprotein sample, we evaluated the PP enrichment and separation efficiency of PNI-*co*-ATBA$_{0.2}$@SiO$_2$ using a dispersive solid phase extraction (dSPE) method[47].

The strong adsorption of PPs on the copolymer surface at high CH$_3$CN content makes it simple to choose appropriate conditions for protein sample loading and removal of NMPs that interfere with the analysis, which is an advantage in terms of enrichment selectivity. Moreover, remarkable difference in binding affinity of SPP and MPPs with the copolymer enables their isolation under different elution conditions (the experimental details are shown in Methods section). By optimizing the rinsing and elution conditions, an efficient protocol via a three-step elution procedure was developed for the enrichment and stepwise isolation of SPPs and MPPs, as illustrated in Fig. 4a. To simulate complex physiological environment, we used semicomplex samples,

including tryptic digests of α-casein mixed with different molar ratios (i.e., 1:10, 1:100 and 1:500) of bovine serum albumin (BSA, a typical non-modified protein widely used in cell culture media) as a model interfering protein. In the case of tryptic digests of α-casein and BSA at a molar ratio of 1:10, PP signals could not be detected before enrichment owing to the severe background interference attributed to the presence of abundant NMPs. After enrichment with our material, most of the NMPs had been removed, and 6 SPPs and 15 MPPs were selectively enriched and effectively divided into two fractions (Supplementary Fig. 49c, d). When the molar ratio of casein to BSA was increased to 1:100, the satisfactory separation capability of the material towards SPPs and MPPs were still maintained. Compared with SPPs, PNI-*co*-ATBA$_{0.2}$@SiO$_2$ showed a much higher enrichment selectivity towards MPPs due to the strong MPP retention and thorough removal of interfering NMPs; up to 15 MPP signals were readily detected in the mass spectrum (Fig. 4b). This high enrichment selectivity was maintained even at the molar ratio of casein to BSA of 1:500 (Fig. 4c), which led to the identification of 12 MPP signals. In sharp contrast, when commercially available TiO$_2$ (GL Sciences, Tokyo, Japan) was evaluated at the molar ratio of casein to BSA of 1:100 under its optimized elution conditions, TiO$_2$ showed strong preference for SPPs, leading to no detectable MPPs in the mass range of 1250–1600 (Supplementary Fig. 50). It is worth mentioning that there have been some improved TiO$_2$ materials displaying satisfactory MPP enrichment capacities in recent years[48–51], the combination of our polymeric material with these improved inorganic materials may be a good choice for practical phosphoproteomics analysis.

**Adsorption capacity and recovery evaluation of the copolymer-based enrichment material**. In addition to the high selectivity towards MPPs, PNI-*co*-ATBA$_{0.2}$@SiO$_2$-based enrichment method exhibited clear advantages in high adsorption capacity and satisfactory recovery towards MPPs, both of which are critical parameters in large-scale and high-throughput proteomics analyses[52]. The adsorption capacity of PNI-*co*-ATBA$_{0.2}$@SiO$_2$ towards MPP was measured by loading different amounts of standard MPPs (i.e., 2pS–4pS) in a fixed volume until the maximal loading amount was reached (Supplementary Fig. 51a–c). When 2pS–4pS were loaded in 85% acetonitrile (ACN)/0.1% formic acid (FA, pH: 2.59), the adsorption capacity of the material increased substantially with the number of phosphates in the MPPs (Fig. 5a), and the adsorption capacities were 30, 106 and 463 mg g$^{-1}$ for 2pS, 3pS and 4pS, respectively. Interestingly, these values sharply decreased to 5 mg g$^{-1}$ when 2pS–4pS were loaded in 75% ACN/0.1% FA (pH: 2.59; Supplementary Fig. 52), which further validated the good tunability of our material by altering solvent polarity. In comparison, the adsorption capacities of aforementioned TiO$_2$ towards 2pS–4pS were much smaller under the optimized loading conditions[39] (Supplementary Fig. 51d–f). The maximal adsorption capacity of TiO$_2$ was only 20 mg g$^{-1}$ towards 4pS.

Importantly, using tri-PPs as examples (i.e., 3pS, 3pT and 3pY), our materials displayed both strong adsorption capacities towards phosphorylation of all three amino acid residues and pronounced difference in adsorption capacity depending on whether the amino acid was serine (106 mg g$^{-1}$), threonine (80 mg g$^{-1}$) or tyrosine (160 mg g$^{-1}$) (Fig. 5b and Supplementary Fig. 53). In comparison, TiO$_2$ displayed weak and unbiased adsorption capacities towards various phosphorylated amino acid residues, which further revealed the good chemoselectivity of our material.

With regard to the recovery of the enrichment method (defined as the ratio of released PP to the total PP, involving in both binding and releasing processes of the PP), TiO$_2$

demonstrated a higher recovery for SPP (e.g., 1pS) than PNI-*co*-ATBA$_{0.2}$@SiO$_2$. However, for MPPs, the recoveries obtained using our material were comparable to (e.g., 2pS) or much higher (e.g., 3pS, 4pS) than those of TiO$_2$ (Fig. 5c). Particularly, the recovery of 4pS reached 90%, which was substantially higher than that obtained using TiO$_2$ (lower than 1%). This could be attributed to the inherent drawback of TiO$_2$ that only a small amount of bound MPPs (i.e., 3pS or 4pS) could be dissociated from the material surface, resulting in low recoveries for 3pS and 4pS. By comparison, owing to strong adsorption capacities towards MPPs and highly controllable binding behaviours of our material, most of the bound MPPs could be disassociated from the copolymer surface, leading to high recoveries for MPPs. Satisfactory recoveries, high adsorption capacities, tunable adsorption behaviours and good chemoselectivity make our material suitable for enrichment and analysis of MPPs.

**MPP enrichment from HeLa S3 cell lysate**. Next, HeLa cell line was used as an example to assess the enrichment performance of PNI-*co*-ATBA$_{0.2}$@SiO$_2$ in real biosamples. Using the optimized enrichment procedure as described in Methods Section, three analytical repeats were performed in parallel and the overlapping between two analytical repeats was 68%. The selectivity of the established method towards phosphopeptides was 81.9%. We identified 2525 unique phosphorylation sites from 1257 unique PPs in 50 μg of HeLa S3 cell lysate (Supplementary Data 1) with high confidence (false discovery rates (FDRs) for protein, peptide and modification sites are all lower than 1%)[53, 54], which was obtained from the pool of three analytical repeats. Interestingly, ~71% of the PPs were multiply phosphorylated species, among which di-PPs, tri-PPs and PPs with four or more phosphorylation sites represented 34, 15.8 and 21.2% of the total identified PP species, respectively (Fig. 5e). By contrast, the use of TiO$_2$ in the previous study led to the conclusion that over 85% of the PPs were singly phosphorylated species[55]. After comparing these data with phosphorylation site information collected from UniProt knowledgebase, we found that 27.7% of the identified phosphorylation sites in SPPs were novel, and the percentage of sites that were novel increased according to the number of phosphates in PPs, to 33.5, 41.3 and 54.1% for di-, tri- and not less than tetraphosphorylation (Fig. 5d), respectively. Considering the HeLa cell lysate is a typical and well-studied cell system and its phosphorylation sites have been fully explored[56], the large amounts of novel multiple phosphorylation sites revealed the strong MPP enrichment capability of our material. It is worth mentioning that some NMPs coeluted with MPPs in the Elution 2 solution. The majority of these NMPs contains multiple adjacent glutamic acids and/or aspartic acids, especially the top 20 NMPs with high signal intensities (their peptide sequences are shown in Supplementary Table 6). Correspondingly, Supplementary Fig. 54 displays the distribution of the signal intensities for the NMPs and PPs identified by our material. Only a few NMPs are more abundant compared with PPs. Among the PPs, the mono-PPs are the most abundant species, followed by di-PPs, tri-PPs and PPs with more than four phosphorylation sites.

Interestingly, PPs captured with our material had larger molecular weights and higher proportions of threonine and tyrosine phosphorylation than theoretical tryptic digests reported in the human protein database and in the literatures. PPs with a molecular weight larger than 3500 Da could be efficiently captured with PNI-*co*-ATBA$_{0.2}$@SiO$_2$, in sharp contrast to negligible PPs in the same mass range obtained using TiO$_2$ (Supplementary Figs. 55 and 56 and Supplementary Table 7).

With regard to the proportions of serine (S), threonine (T) to tyrosine (Y) phosphorylation identified with our material, the ratios of S:T:Y phosphorylation (86.6%:10.0%:3.4%) for SPPs were close to the data reported in many previous studies[57]. However, the ratios of threonine and tyrosine phosphorylation increased remarkably when MPPs were accounted for. For example, the ratio of S:T:Y reached 60.0%:21.7%:18.3% or 51.9%:35.4%:12.7% for tri-PPs, or not less than tetra-PPs, respectively (Fig. 5f), which suggested that T and Y phosphorylation might be more prevalent in MPPs (Supplementary Table 8). The powerful enrichment capacities of our material towards threonine and tyrosine MPPs may facilitate the identification of more T or Y phosphorylation sites and promote the exploration of their unique regulation effects.

## Discussion

In summary, above results reveal four attracting features of our material: specific and tunable complexation with MPPs driven by multiple H-bonding interaction, smart adsorption conversion windows modulated by solvent polarity, solution pH or temperature, high adsorption capacities and recovery, as well as extraordinary enrichment capacities towards diverse MPPs (particularly the rare tyrosine MPPs) from real biosamples. Benefiting from these features that are fundamentally different from conventional MOAC or IMAC materials, large amounts of unique multiple phosphorylation sites have been identified from HeLa cell lysate. High discovery rates for unique phosphorylation sites and high proportions of threonine and tyrosine phosphorylation for MPPs, illustrate the great potential of our material in comprehensive phosphoproteomics analysis. Therefore, studies of protein multisite phosphorylation may be largely promoted by the convergence of this intelligent enrichment material with MS-based analysis method[58].

Furthermore, the smart polymer-based design idea could be expanded to other protein PTMs, such as methylation, acetylation, hydroxylation and ubiquitination, which have unique regulatory effects similar to that of phosphorylation[59], but these PTM peptides cannot be efficiently captured by artificial materials by far. Through precise design of recognition units for these PTM sites and dynamic control of PTM-peptide adsorption behaviours on material surfaces, these knotty enrichment challenges might be tackled by smart polymer-based materials and methods. We anticipate that PTM proteomes may develop into a promising application direction for smart polymers in the near future[60].

## Methods

**Synthesis and characterization of ATBA monomer.** Triethylamine (0.505 g, 5 mmol) was added to a solution of 4-thioureido-benzoic acid (0.98 g, 5 mmol) in 30 mL dry chloroform, the mixture was stirred for 10 min, then acryloyl chloride (0.453 g, 5 mmol) was added dropwise to this mixture at ambient temperature, and the mixture was stirred for 24 h (Supplementary Fig. 1). After that the mixture was washed with water three times, and the organic layer was dried over anhydrous sodium sulphate overnight. After filtration and evaporation of solvent, the crude product was purified by column chromatography on silica gel with an elution of dichloromethane/methanol ($CH_2Cl_2/CH_3OH$, v v$^{-1}$: 2:1), to obtain the pure product as yellow powder (1.07 g, yield: 78%, m.p. 161 °C). $^1$H NMR (300 MHz, $CDCl_3$): δ (p.p.m.): 6.18 (d, J = 6 Hz, 1H, C=C*H*), 6.62 (d, J = 9 Hz, 1H, C=C*H*), 6.77–6.81 (m, 1H, C=C*H*), 8.01 (d, J = 6 Hz, 2H, Ph-*H*), 8.11 (d, J = 6 Hz, 2H, Ph-*H*), 11.91(s, 1H, CN*H*CS), 12.94 (s, 1H, CN*H*CS), 13.13(s, 1H, COO*H*); $^{13}$C NMR (300 MHz, $CDCl_3$): δ (p.p.m.): 126.5, 128.7, 131.3, 135.4, 136.0, 157.6, 167.2, 172.4, 175.9 (Supplementary Fig. 2a, b); IR spectroscopy: 3183, 2972, 2851, 1673, 1599, 1570, 1410, 1252, 1165, 978, 875 cm$^{-1}$; matrix-assisted laser desorption ionization MS: m/z calcd. for $C_{11}H_{10}N_2O_3S$: 250.04; found: 251.0627 (M+H)$^+$ (Supplementary Fig. 2c). Elemental analysis calcd. (%) for $C_{11}H_{10}N_2O_3S$: C, 52.79; H, 4.03; N, 11.19; S, 12.81. Found: C, 52.88; H, 4.00; N, 11.23; S, 12.76.

### Preparation of PNI-*co*-ATBA$_{0.2}$-modified silica gels.

(a) Silica gels (4.0 g, average particle size: ~5.0 µm, average inner pore diameter: ~300 Å) was suspended in 25 mL of hydrochloric acid (HCl, 0.1 mol L$^{-1}$) for 48 h at ambient temperature, to generate sufficient hydroxyl groups on the silica surface. The obtained silica gels were filtered and washed with water and then ethanol three times, which was then dried under vacuum.

(b) Aminopropyl-trimethoxysilane (4.0 mL) was dissolved in anhydrous toluene (40 mL), and the aforementioned silica gels (4.0 g) were added. The mixture was stirred and refluxed for 6 h. The product was separated by centrifugation at 7000 r.p.m. for 5 min. Then, the amino-modified silica gels were washed three times with toluene and then ethanol by repetitive dispersion/ precipitation cycles to remove the unreacted materials, and then the silica gels were dried under vacuum.

(c) The amino-modified silica gels (4.0 g) were suspended in 30 mL of anhydrous $CH_2Cl_2$ with pyridine (0.8 mL). The polymerization initiator bromoisobutyryl bromide (BIBB, 4.0 mL) was added dropwise to this solution for 30 min at 0 °C, which was continued to stir overnight at ambient temperature. (This reaction should be protected from light.) The product was separated by centrifugation at 7000 r.p.m. for 5 min. The silica gels were washed six times with dry $CH_2Cl_2$ by repetitive dispersion/precipitation cycles, and were then dried under vacuum.

(d) PNI-*co*-ATBA was grafted onto silica gels through a surface initiated ATRP. NIPAAm (2.26 g), ATBA (1.00 g) (20 mol% ATBA against NIPAAm) was dissolved in a mixture of $H_2O$ (10.0 mL), methanol (10.0 mL) and N, N'-dimethyl formamide (2.0 mL), then the aforementioned BIBB-modified silica gels (4.0 g) were added. The solution was further deoxygenated by three freeze-pump-thaw cycles, and then copper bromide (0.0567 g, 0.4 mmol) and N,N,N',N',N''-pentamethyl-diethylenetriamine (0.0693 g) were added. The solution was continued to stir for 6 h at 60 °C. The product was separated by centrifugation at 7000 r.p.m. for 8 min. The resulting copolymer-modified silica gels were washed three times with distilled water and then methanol by repetitive dispersion/precipitation cycles to remove the unreacted materials. PNI-*co*-ATBA$_{0.2}$ modified silica gels (denoted as PNI-*co*-ATBA$_{0.2}$@SiO$_2$) was thus obtained and dried under vacuum. The detailed characterization data is shown in Supplementary Fig. 48.

**Trypsin digestion of proteins.** For digestion standard proteins, 1 mg protein was dissolved with 100 µL of 6 M urea in 50 mmol L$^{-1}$ (abbreviated to mM) ammonium bicarbonate (NH$_4$HCO$_3$). Then, 2 µL 50 mM DL-dithiothreitol (DTT) were added to the protein solution and the resulting solution were stored at 56 °C for 45 min. After adding 5 µL 50 mM iodoacetamide (IAA) the mixture was incubated in the dark for 30 min at room temperature. Finally, trypsin was used to digest the protein at an enzyme to protein ratio of 1:30. The digestion was stopped with 5 µL FA (10%) after 14 h.

For digestion of HeLa S3 cell lysate, 1 mg protein was taken with the addition of 2 µL 50 mM DTT and 5 µL 50 mM IAA. After keeping in the dark for 1 h, the protein was centrifuged at 14,000 r.p.m. for 30 min. After being washed with 100 µL NH$_4$HCO$_3$ (50 mM) twice, 25 µg trypsin dissolved in 200 µL NH$_4$HCO$_3$ was mixed. After digestion for 16 h, digests were centrifuged at 14,000 r.p.m. for 40 min and filtrate was collected.

**Cell culture and protein extraction.** HeLa S3 cells were cultured in RPMI-1640 with 10% fetal bovine serum, 4.5 g L$^{-1}$ glucose, 2 mM glutamine and 100 µg mL$^{-1}$ penicillin/streptomycin at 37 °C with 5% $CO_2$. The native cells were harvested when they reached 90% confluence in T-75 flasks. To extract protein from the cells, the cells were centrifuged at 1000 r.p.m. and washed with phosphate buffer solution twice. The cells were mixed with precooled lysis buffer (50 mM Tris-HCl, pH = 7.4, 8.0 M urea, 65 mM DTT, 1 mM ethylene diamine tetraacetic acid, 1% (v v$^{-1}$) protease inhibitor cocktail, 1% (v v$^{-1}$) Triton X-100, 1% (v v$^{-1}$) phosphatase inhibitor cocktail 3). Phosphatase inhibitor cocktail 3 contains cantharidin, (-)-p-bromolevamisole oxalate and calyculin A. Protease inhibitor cocktail contains individual components including 4-(2-aminoethyl)benzenesulphonyl fluoride hydrochloride, aprotinin, bestatin, E-64, leupeptin and pepstatin A. Then, cells were treated with sonication. The resulting solution was transferred to centrifuge tube and centrifuged at 15,000 r.p.m. for 30 min at 4 °C. The supernatants were collected and stored at −80 °C for further use.

**Enrichment protocols of PPs with PNI-*co*-ATBA$_{0.2}$@SiO$_2$, TiO$_2$ and IMAC.** PP enrichment was carried out in a dSPE mode[47]. The model sample was tryptic digest of α-casein with different folds of BSA interference. For the case of enrichment of PPs with PNI-*co*-ATBA@SiO$_2$ (silica gel: diameter: 5 µm and inner pore size: 300 Å), 1.0 mg polymeric material was suspended in $CH_3CN$ and loaded into an Eppendorf tube. After conditioning and equilibrating with 100 µL of 50% $CH_3CN$/0.1% trifluoroacetic acid (TFA, pH: 1.95) and 75% $CH_3CN$/0.1% FA (pH: 2.59), 100 µL tryptic α-casein digests (200 fmol µL$^{-1}$) redissolved with 75% $CH_3CN$/0.1% FA (pH: 2.59) were incubated with PNI-*co*-ATBA$_{0.2}$@SiO$_2$ materials. After incubation for 30 min, the slurry was centrifuged and supernatant was discarded. The remaining PNI-*co*-ATBA@SiO$_2$ materials were rinsed twice with 100 µL of 70% $CH_3CN$/0.1% FA (pH: 2.59). Subsequently, the trapped mono-PPs and multi-PPs were eluted twice with 50 µL of 50% $CH_3CN$/0.3% FA (pH: 2.34) and 50% $CH_3CN$/3% TFA (pH: 0.86), respectively, after being shaken for 2 min.

With the increased levels of BSA interference, the rinse volumes were increased accordingly, for casein:BSA of 1:100 or 1:500, the rinse volume was 100 μL × 2 and 100 μL × 4, respectively. The eluted peptides were characterized with a nano electrospray ionization tandem quadrupole/orthogonal-acceleration time of flight mass spectrometer (ESI-Q-TOF MS). The detailed mass spectra are shown in Supplementary Fig. 49.

For the PP enrichment from HeLa S3 cell lysate, the amount of PNI-co-ATBA$_{0.2}$@SiO$_2$ materials was adjusted to 2.5 mg when PPs were enriched from 50 μg real complex samples. After sample loading, PNI-co-ATBA$_{0.2}$@SiO$_2$ materials were rinsed with 100 μL × 4 of 70% CH$_3$CN/0.1% FA (pH 2.59) and 50 μL × 2 of 50% CH$_3$CN/0.3% FA (pH 2.34). The bound MPPs on PNI-co-ATBA$_{0.2}$@SiO$_2$ were eluted with 50 μL of 50% CH$_3$CN/3% TFA (pH 0.86, Elution 2), a few NMPs (Supplementary Table 6 and Supplementary Data 2) were coeluted with MPPs.

Enrichment of PPs with commerically available TiO$_2$ was based on previous reports with minor modifications[39]. Briefly, TiO$_2$ material (2.0 mg) was slurried in CH$_3$CN and pushed into a GE-Loader tip. The microtip was conditioned and equilibrated with 40 μL of loading buffer (i.e., 80% CH$_3$CN containing glycolic acid (1 mol L$^{-1}$) and TFA (volume ratio: 5%)) and washing buffer (80% CH$_3$CN containing 1% TFA, pH = 1.07), respectively. The tryptic peptide mixture was dried, redissolved in 20 μL loading buffer and loaded onto the TiO$_2$ microcolumn. The bound peptides were washed with 40 μL of loading buffer and 40 μL washing buffer, and eluted with 40 μL of 0.1% ammonium hydroxide (pH: 10.82). The elution fraction was immediately acidified with 1% FA. The procedure for PP enrichment with IMAC (e.g., Fe$^{3+}$) was the same as previous reports[52]. IMAC material (1–2 mg) was pushed into a GE-Loader tip. The microtip was conditioned and equilibrated with 40 μL of loading buffer (50% CH$_3$CN/0.1% TFA, pH: 1.95), respectively. The tryptic peptide mixture was dried, redissolved in 20 μL loading buffer and loaded onto the IMAC microcolumn. The bound peptides were washed with 40 μL of loading buffer and 40 μL of 20% CH$_3$CN/1% TFA (pH: 1.07), and eluted with 40 μL of 0.5% ammonium hydroxide (pH: 11.20). The elution fraction was immediately acidified with 1% FA (pH: 2.15). The detailed mass spectra of PPs enriched with TiO$_2$ and IMAC is shown in Supplementary Fig. 50.

**Determination of adsorption capacity towards PPs**. To measure the adsorption capacity of PNI-co-ATBA@SiO$_2$ toward PPs, fixed concentration of serine di-, tri- and tetra-PPs (2pS–4pS) at different volume were dissolved in 85% ACN/0.1% FA (pH: 2.59) or 75% CH$_3$CN/0.1% FA (pH: 2.59) and loaded into 1 mg materials packed GE-Loader tips, respectively. The flow through was collected and analysed with Q-TOF MS. The adsorption capacity was measured as the highest PP to adsorbing material ratio at which no PP signals were observed in the MS spectra. The experiments were repeated once. The detailed figures are shown in Supplementary Figs. 51a–c, 52 and 53.

To measure the adsorption capacity of TiO$_2$ toward PPs, fixed concentration of 2pS–4pS at different volume were dissolved in loading buffer (80% CH$_3$CN containing 1 M glycolic acid and 5% TFA) and loaded into 1 mg materials packed GE-Loader tips, respectively. The flow through was collected, dried and dissolved in 50% CH$_3$CN/0.1% FA (pH: 2.59). Then, the solution was analysed with Q-TOF MS. The adsorption capacity was measured as the highest PP to adsorbing material ratio at which no PP signals were observed in the MS spectra. The experiments were repeated once. The detailed figures are shown in Supplementary Fig. 51d–f.

**Determination of enrichment recovery of PPs**. 1pS–4pS were used as model peptide sample to determine the enrichment recovery of both PNI-co-ATBA$_{0.2}$@SiO$_2$ and TiO$_2$-based method. Samples before and after enrichment were analysed with MS. The enrichment recovery was calculated based on the MS intensity ratio of the model peptide (1pS–4pS) after and before enrichment.

**Nano ESI-MS and nano liquid chromatography-mass spectrometry analysis**. PP enriched from α-casein digests (Supplementary Table 5) were analysed with a nano ESI Q-TOF MS (Waters, Milford, MA, USA). The samples were infused into the ESI source with Nano Acquity UPLC (Waters). The MS analysis was performed under the positive ion mode. Capillary voltage was 2.3 kV and source temperature was 100 °C. Full scan MS data and tandem MS/MS data were acquired at m/z 600–2000 and 100–2000, respectively.

The PPs obtained from HeLa S3 cell lysate were separated and characterized using LTQ-Orbitrap Velos coupled with Accela 600 HPLC System (Thermo, CA, USA). For the reverse-phase liquid chromatography separation, 0.1% FA (pH: 2.59) in water and in CH$_3$CN was used as mobile phases A and B, respectively. The analytical column with an inner diameter of 75 μm was packed in-house with Daisogel C18 AQ particles (3 μm, 120 Å) to 12 cm length. The flow rate was set at 200 nL min$^{-1}$. The 180 min gradient elution was performed with a gradient of 0–2% B in 2 min, 2–35% B in 150 min, 35–80% B in 6 min, 80% B in 6 min, 80% B–100% A in 1 min and 100% A for 15 min. Full mass scans were carried out on the Orbitrap Velos analyser with acquisition range from m/z 400 to 2000 (R = 60,000 at m/z 400). The 20 most intense ions from the full scan were selected for fragmentation via collision induced dissociation (CID) in the ion trap (relative collision energy for CID was set to 35%). The dynamic exclusion function was set as follows: repeat count 1, repeat duration 30 s and exclusion duration of 60 s.

**MS data analysis**. All the RAW data files obtained from Xcalibur 2.1 were converted into *.mgf file format using Proteome Discoverer 1.2 (Thermo Scientific). Then.mgf files were searched using the Maxquant (version 1.3.0.5)[54] against Uniprot protein fasta database of human. Carbamidomethylation on cysteine was set as a fixed modification, whereas oxidation on methionine, acetylation of protein N-term and phosphorylation on serine, threonine and tyrosine were set as the variable modifications. Trypsin was set as the specific proteolytic enzyme with up to two missed cleavages allowed. Peptides with charge states of 2, 3 and 4 were chosen for further fragmentation. The mass tolerance for the precursor ion was set to 10 p.p.m. and MS/MS tolerance was 0.8 Da for LTQ-Orbitrap Velos. The FDR for protein, peptide and phosphorylation modification sites are all set as <1%. The minimum peptide length was set as 6 and other parameters for database searching were default values in Maxquant. The obtained phosphorylated sites were further filtered by localization probability > 0.5. Moreover, the data were searched against reverse and contaminant sequences. PP with the same sequence and phosphosites but having other modifications (e.g., oxidation on methionine residue) were taken as one PP peptides with the same amino acid sequence, but different phosphorylation states were identified in different PPs.

**Data availability**. The authors declare that (the/all other) data supporting the findings of this study are available within the paper (and its Supplementary Information Files).

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

## Acknowledgements

This work was supported by the Major State Basic Research Development Program of China (973 Program) (2013CB933002), China National Funds for Distinguished Young Scientists (51325302), the National Natural Science Foundation of China (51533007, 21475129, 51521001, 21275114, 51473131, U1508221), Program for Changjiang Scholars and Innovative Research Team in University (IRT_15R52). G.Q. acknowledges Hubei Provincial Department of Education for financial assistance through the Chutian Scholar Program and Hubei Provincial Natural Science Foundation of China (2014CFA039).

## Author contributions

G.Q. and Q.L. contribute equally to this work. G.Q. designed the smart copolymer, and was in charge of experiments for mechanistic studies; Q.L. synthesized the copolymers and performed the adsorption experiment; X.-L.L. evaluated the performance of materials in phosphopeptide enrichment; J.L. aided X.-L.L. in experiment and data analysis; M.Y. provided the guidance on the PP enrichment from HeLa cell lysate. T.S. proposed the ideas and wrote the paper; T.S. and X.-M.L. designed the whole work and analysed all the results. All authors commented on the final draft of the manuscript and contributed to the analysis and interpretation of the data.

## Additional information

**Competing interests:** The authors declare no competing financial interests.

