## [Peer Review File · Nature Communications]

Reviewers' comments:

Reviewer #1 (Remarks to the Author):

The manuscript titled "Hydrogen bond-based smart polymer for highly selective and tunable capture of multiply phosphorylated peptides" presents an exciting new technology that addresses an unmet need in the field of phosphoproteomics. The authors provide evidence that support their claim that the guest molecule in the polymer and the intact smart co-polymer has a higher affinity for multiply phosphorylated peptides than for singly phosphorylated peptides and non-phosphorylated peptides under a set of well characterized conditions. The authors also provide evidence that support their claim that they have determined conditions for eluting the multiply phosphorylated peptides that allow them to recover the majority of all of the multiply phosphorylated peptide in their original samples (Table S5). Additionally the authors somewhat address the purity of their eluted samples in Figure S38. Specifically, it is exciting to see the very low number of non-modified peptides measured in elution 2 in the 1:100 casein: BSA sample. Lastly, it is exciting that the paper introduces the use of smart polymers and hydrogen bonding as novel strategies for phosphopeptide enrichment that are complementary to chelation interaction based methods. We recommend that the paper is accepted contingent on addressing our major points outlined below.

(1) The plots of the mass spectrometry spectra in Figure S38 sufficiently address concerns about purity of the samples after the two elutions. However, this issue is not addressed in the context of the HeLa cell extract experiment. The authors should report the non-phosphorylated peptides that they measure in that experiment in Table S6 or they should explicitly state that they do not identify any such peptides. They should also make clear whether these peptides were measured in Elution 1 or 2. Additionally, a figure showing the distribution of the signal intensities for the peptides with different numbers of phosphorylations (including 0) should be included. Some of the phosphorylated peptide signal values included in the table are 0 and thus it is unclear why they are included. In Figure 5e the portion of nonmodified peptides should also be reported.

The degree of contamination with non-modified peptides should be addressed (i.e. purity) since the number of non-phosphorylated peptides in complex samples is so much higher than phosphorylated ones, large differences in affinity alone do not guarantee that a substantial amount of nonphosphorylated peptides will not be in the post-enrichment sample. Knowing the number of nonmodified peptides identified and their intensity is important for judging the significance of this paper because mass spectrometry is not able to identify and measure all of the peptides that are in a sample and the peptides that are identified and measured are generally the ones that are more abundant. This bias to measure the more abundant peptides rather than the less abundant peptides only increases in multiplexed mass spectrometry. While this paper did not address multiplexed mass spectrometry, it is an increasingly important tool for interpreting protein phosphorylation in their physiological contexts and thus the compatibility of the reported smart polymer based enrichment method with this mass spectrometry method needs to be addressed.

On a related note, there is supplemental data in Figure S38 and Figure S39 that more clearly demonstrates the strengths of this method than what is currently shown in Figure 4. Specifically, panels C and D from Figure S38 could be shown along with panels E and F from Figure S39. The results from the 1:10 and 1:500 mixture experiments with PNI-co-ATBA @ SiO₂ currently in Figure 4 could be put instead in Figure S38. The authors could also choose a different color for labeling the 3P and 4P peptides in order to make the benefits of their method more clear.

(2) The first paragraph of the introduction (lines 26-38) is unacceptably weak and confusing. The authors do not make clear the distinction between multiple phosphorylations on a protein that would be seen on separate peptides and multiple phosphorylation sites where those phosphorylations would be found on the same peptide. The authors use retinoblastoma protein and Sic1 as examples where we know about the regulatory importance of multiple phosphorylations on a same peptide but the explanation of the "Rb phosphorylation codes" and the

6 sites of Sic1 that are phosphorylated are not explicit enough to make clear the importance of being able to measure single peptides that have multiple phosphorylations on them. The authors could also consider including a schematic of a protein with multiple phosphorylations very close to each other on sequence space as a first panel of Figure 1.

(3) Clearly not all possible peptide sequences can be tested with model peptides, however, examination of the sequences of the peptides that the authors raises two concerns that should be mentioned in the text. Firstly, the sequences all include four prolines, raising the concern that their results might not apply to phosphorylations that are not proline directed. The authors should comment on the bias this sequence might represent. Secondly, the model acidic peptide NMP2 includes three aspartic acids but they are all separated in the peptide sequence. A quick scan of the peptide sequences reported in Table S6 shows that many of the phosphorylated peptides also include regions of poly (E) and poly (D). Readers who have done phosphoproteomics will know that many of the contaminating acidic peptide sequence include stretches of glutamic and aspartic acid residues right next to each other. The bias that their choice of acidic peptide introduces should be mentioned and it would be ideal to include a supplemental figure that shows data about the distribution of contaminating acidic peptide sequences that are seen in mass spectrometry during enrichment for phosphorylated peptides.

(4) The authors need to make clear their definition of the word "site" and the degree to which they are considering redundancy between peptides when quantifying the percentage of new sites and proportions of different peptides found in Figure 5. Inspection of Table S6 shows that the authors are calling each of the three phosphorylations on a tri-PP a different site, but this is not clearly stated in the paper. Additionally, inspection of the Table S4 shows that often missed cleavages of lysines on peptides or oxidations of methionine introduce modifications in the peptide sequence that do not give more biological information about phosphorylations. (Peptides No.1 and No. 4 in Table S4 are an example of this.) If the authors are counting these peptides as two different di-peptides then they are misrepresenting the presence of biological meaningful diversity in the multi-PPs that they are recovering.

Minor Comments

Figures Figure 1.

Panel h. There should be an upward arrow before pH and T because some of the audience might not be as familiar with polymer physics as the authors.

Line 399. The method used to measure the contact angle in panel E is not mentioned in the figure legend or the associated results section.

Figure 2.

Panels j, h. The plots should have the same Y-axis as the plots in a, b, d, e, g, and h. The plots should also have the same x-axis as the plots above or the reason why the experiments were done for a shorter amount of time should be stated.

Figure 3.

Panel a, b. Why is there no acetonitrile in the solution? Is that a typo or a constraint due to the AFM. If there is not acetonitrile in the solution it should be mentioned in the text since the importance of acetonitrile concentration is discussed extensively.

Line 433. Is the use of the word "elution" a typo that should actually read "solution". Elution does not make sense in the context of this figure.

Figure 4.

Panel a. The schematic for the enrichment and strategy should include the solution pHs and the temperature(s) at which the experiments are performed.

Panels b, c, d. It appears that the Y-axis of Relative Abundance was determined by normalizing the signals by the largest signal associated with a phosphorylated peptide. If this is the case that should be stated, and if not the normalization method should be noted.

Figure 5.

Panel c. It would be great if the Recovery experiment could be done with the NMP1 and NMP2.

Panel d. The Y-axis should go to 100%.

Panels c, d, f. Since the number of repeats for the experiment is not too large the data points themselves should be plotted (x-scatter can be added so that separate points resolved) rather than the bar-plot with the standard deviation bars.

Panel e. The data should be represented with either a bar plot or table, but not with a pie chart. The fraction of non-phosphorylated peptides should also be included.

Text

Line 64-65. Hydrogen bonds are often considered a type of electrostatic interaction to it is confusion to contrast H-bonding to both chelation and electrostatic interactions.

Line 72. This sentence is confusing because it is not clear whether the molecules referred to in "biding or release of these molecules" are the same as the "guest molecules" mentioned earlier in the sentence.

Line 75. The sentence on this line can be stronger. The paper does show that their ATBA copolymer is an efficient tool to modulate these behaviors and thus the "may" does not seem appropriate.

Reviewer Comments on the Supplemental Materials of Qing et al.

There are multiple times in the supplemental methods where the authors talk about washing a material a certain number of times and use the grammar "after washing for X times". It should be "after washing X times".

343-344 Can the authors give more information about "protease inhibitor cocktail" and "phosphatase inhibibtor cocktail (?3)

Supplemental methods 2.15 - While the authors do give the concentrations of strong acids it would be good to also include measured pH of the solutions as well.

361-362 - What were the rinse volumes increased to for the increased levels of BSA interference?

365-366 - The polymer to peptide mass ratio will be important to users of the method. Can the authors give more specifics about this besides that "the material was adjusted slightly to 2-3 mg" For the enrichment of PPs with commercially available TiO₂, what was the peptide starting material : enrichment material ratio?

Line 392 (and 398/399) the statement "The adsorption capacity was measured before the point that the PP signals were observed" is unclear. I would suggest saying, "the adsorption capacity was measured as the highest PP to adsorbing material ratio at which no PP signals were observed in the MS spectra".

406 - The meaning of the word "targeted" is unclear.

483-484 It is unclear if the authors are talking about the work that the results of Figure S2 inspired them to do or if they are talking about how they are interpreting the results of Figure S2.

Figure S5, S6 - What is [G]/[H] stand for? What is Log[G]?

Figure S9. The pHs could be added to the inside right corner of each of the plots to make the figure easier to interpret.

Figure S14. it is unclear whether buffer was not used for both the experiment at different pHs or only the experiment where the peptides were also added. If there were no buffer for the measurements at different pHs, how was the pH maintained?

Figure S25. Should it be that different pH-mediated adsorption conversion windows were seen for the 3pY and 3pT? It is not clear why it is reported as two different windows for 3pY.

Figure S30. The differences in surface morphology should be quantified if you are going to draw conclusions from them.

Figure S43. The purpose of this figure is not clear and it is not clear what is meant by "new information".

Figure S45. This should be a table not a bar chart. There is no information added by showing the values as bars.

Figure S47. This data should be represented in a table

Reviewer #2 (Remarks to the Author):

Guangyan Qing et al propose a new enrichment approach for multiply phosphorylated peptides based on smart polymer. Overall, this is a novel and interesting new approach with broad applications in the field of proteomics. The development and characterization of the material is well described. Probably that section of the manuscript could be shorter and instead the information provided in the supplementary section.

The key to this type of article is whether the technique performs well when dealing with real biological samples. The authors use a HeLa cell lysate to demonstrate the performance of their approach. This section of the manuscript is short. The following issues would need to be addressed:

- How many biological and analytical repeats were done? This seems to be a single analysis.
- When I analyzed table S6, it seems to me that phosphopeptides were reported more than once. When I compared the modified sequences, I only saw 695 unique phosphopeptides. The rest of the entries were repeats of the same sequences. Furthermore, 26 of the remaining 695 unique phosphopeptides had identical m/z and MS/MS IDs. This would mean that only 669 unique phosphopeptides are present in this table instead of the 1168 reported in the text. The authors need to address this issue.
- of the remaining 669 unique phosphopeptides, 25 had charges of 4+ or more. Considering that the mass tolerance for the MS/MS was 0.8Da, I am not confident that these matches are right.
- of the remaining 669 unique peptides, 60 have mass errors on the parent ion > 1.5 ppm with some with over 5ppm. These seem to be outliers and the authors should verify that this is right.
- The fact that the MS/MS were done in the trap could be problematic. In particular, considering that the tolerance on the MS/MS ions is 0.8 Da. It seems to me that the confidence would be much higher if the authors had performed MS and MS/MS at high resolution. This would be particularly the case for phosphopeptides with multiple phosphorylation sites.
- The distribution of the Mascot score appears to be dependent on the number of phosphosites. Peptides with the highest number of phosphosites seem to have the lowest Mascot score. The author applied a Mascot score cutoff of 30. However, I am not sure that this is appropriate for phosphopeptides that have multiple phosphorylation sites.
- The method section for the database search and filtering need to be improved.
- Thank you for providing a few examples of the annotated MS/MS. They should all be provided for the unique phosphopeptides. It is likely that some of the MS/MS are from overlapping peptides. This would explain why some of the large ions in MS/MS are not annotated. The authors need to discuss this issue and its impact on the results considering the 0.8Da mass tolerance in MS/MS.
- The authors need to use other tools to validate their results and in particular the position of their sites. See Nat Biotechnol. 2013 Jun; 31(6):557-64. doi: 10.1038/nbt.2585. Epub 2013 May 19.

Reviewer #3 (Remarks to the Author):

To aim at specifically enriching multiple phosphorylated peptides (MPPS), authors synthesized a thermal-responsive polymer decorated with polar functional groups (ATBA) that are designed to interact with phosphorylated peptides through hydrogen bonding. Although many techniques, such as IMAC, MOAC, strong cation exchange chromatography (SCX), strong anion exchange chromatography (SAX), and HILIC, have been well developed for the enrichment of phosphorylated peptides, authors state that their new method has two main advantages: (i) the method is based on hydrogen-bonding and has highly specific and efficient enrichment of MPPS, and (ii) the performance of their method is tunable.

Techniques that are based on specific affinity binding, e.g., MOAC and IMAC, are generally believed to have good specificity and efficiency for enriching phosphorylated peptides. Methods that rely on weak hydrogen bond, e.g., HILIC, are usually less specific and less efficient for enrichment of phosphorylated peptides. The results described in this study are contrary to the established understanding. Therefore, the authors must provide strong evidence and reasonable explanations to support their conclusions.

Overall, the experiments and results currently described in this paper are not sufficient to support the conclusions. When the advantages of the new method are clearly demonstrated and reasonably understood, a novel application of the method to studying a phosphorylation system of biological significance would strengthen the manuscript.

1. Modulation of binding of phosphorylated peptide by using solution conditions, such as solvent polarity and pH, is not unique to this method. Such modulation by changing solution conditions can also be achieved using other enrichment methods.

2. Authors reasoned that high specificity and efficiency of the method for enriching MPPS result from involvement of multiple complementary H-bonds. However, authors did not provide K_a values of interactions between the polymer and different model peptides. The K_a values provided on the interactions between monomer ATBA and different model peptides cannot represent interactions involving multiple phosphate groups of a single peptide and multiple ATBA groups of a single polymer.

3. Polymers having different contents of ATBA should be used to study their binding with different peptides, which would reveal how the density of ATBA groups affects the interaction.

4. Because ATBA is randomly distributed in the linear polymer and phosphate groups are also present at different locations in the peptides, how specific can this method achieve the binding of different peptides containing multiple phosphate groups at different amino acid sites?

5. Page 4, line 88-93, the K_a of ATBA monomer to hydrogen phosphate (K_a : $5.34 \times 10(4)$ L/mol) was only seven times higher than that with acetate (main source of acidic peptide, K_a : $7.6 \times 10(3)$ L/mol). Considering that two amino acids contain acetate groups in their side chain, the concentration of peptides containing multiple acetate groups could be significantly higher than that of MPP. Therefore, how does such small difference in binding affinity allow for differentiating MPP from peptides containing multiple acetate groups? Based on such difference in K_a value, how will authors explain their results that the polymer is able to enrich PPs from 1:500 ratio of BSA?

6. Results in Figure 4e and Figure 5c show very poor or no interaction of 4pS with TiO_2 . However, the enrichment of MPP using TiO_2 -based materials have been well demonstrated by many groups. (Wan, J.J., et al., TiO_2 -Modified Macroporous Silica Foams for Advanced Enrichment of Multi-Phosphorylated Peptides. *Chemistry-a European Journal*, 2009. 15(11): p. 2504-2508. Yue, X.S., A. Schunter, and A.B. Hummon, Comparing Multistep Immobilized Metal Affinity Chromatography and Multistep TiO_2 Methods for Phosphopeptide Enrichment. *Analytical Chemistry*, 2015. 87(17): p. 8837-8844. Zhao, X.Y., et al., Citric Acid-Assisted Two-Step Enrichment with TiO_2 Enhances the Separation of Multi- and Monophosphorylated Peptides and Increases Phosphoprotein Profiling. *Journal of Proteome Research*, 2013. 12(6): p. 2467-2476.)

7. Page 6, line 134, TiO_2 or ZrO_2 have much weaker adsorption of PPs. Why is the recovery of 1-3Ps from the use of TiO_2 and polymer comparable (Figure 5C)?

8. As shown, in the fluorescence titration experiment (Figure S5), up to 150 μ M of H_3PO_4 , $H_2PO_4^-$, HPO_4^{2-} and PO_4^{3-} were added into 10 mM TrisHCl (pH 7.4). With the Tris buffer at pH 7.4, the composition of phosphate ions should be similar. How would authors explain the differences in K_a ?

9. In Figure 5, why is the recovery of 4pS so low on TiO_2 ? Does this mean that TiO_2 cannot capture 4pS, but can capture 1pS, 2pS and 3pS? However, the adsorption capacity of TiO_2 for 4pS is higher than for 1pS, 2pS and 3pS (Figure 5a). How would authors explain this inconsistency?

10. A significant application should be demonstrated, such as analysis of samples with clinical/biological significance.

Referee 1:

The manuscript titled “Hydrogen bond-based smart polymer for highly selective and tunable capture of multiply phosphorylated peptides” presents an exciting new technology that addresses an unmet need in the field of phosphoproteomics. The authors provide evidence that support their claim that the guest molecule in the polymer and the intact smart co-polymer has a higher affinity for multiply phosphorylated peptides than for singly phosphorylated peptides and non-modified peptides under a set of well characterized conditions. The authors also provide evidence that support their claim that they have determined conditions for eluting the multiply phosphorylated peptides that allow them to recover the majority of all of the multiply phosphorylated peptide in their original samples (Table S5). Additionally the authors somewhat address the purity of their eluted samples in Figure S38.

Specifically, it is exciting to see the very low number of non-modified peptides measured in elution 2 in the 1:100 casein: BSA sample. Lastly, it is exciting that the paper introduces the use of smart polymers and hydrogen bonding as novel strategies for phosphopeptide enrichment that are complementary to chelation interaction based methods. We recommend that the paper is accepted contingent on addressing our major points outlined below.

Reply: Thanks for your comments, which will encourage us to use the material tool for resolving the challenges in PTM-proteomics. We appreciate your professional suggestions and careful corrections. According to your questions, we add some new experimental data and revise the manuscript and the supplementary materials. The detailed reply is listed as below:

Question 1a: The plots of the mass spectrometry spectra in Figure S38 sufficiently address concerns about purity of the samples after the two elutions. However, this issue is not addressed in the context of the HeLa cell extract experiment. The authors should report the non-phosphorylated peptides that they measure in that experiment in Table S6 or they should explicitly state that they do not identify any such peptides.

They should also make clear whether these peptides were measured in Elution 1 or 2. Additionally, a figure showing the distribution of the signal intensities for the peptides with different numbers of phosphorylations (including 0) should be included. Some of the phosphorylated peptide signal values included in the table are 0 and thus it is unclear why they are included.

In Figure 5e the portion of non-modified peptides should also be reported. The degree of contamination with non-modified peptides should be addressed (i.e. purity) since the number of non-phosphorylated peptides in complex samples is so much higher than phosphorylated ones, large differences in affinity alone do not guarantee that a substantial amount of nonphosphorylated peptides will not be in the post-enrichment sample. Knowing the number of nonmodified peptides identified and their intensity is important for judging the significance of this paper because mass spectrometry is not able to identify and measure all of the peptides that are in a sample and the peptides that are identified and measured are generally the ones that are more abundant. This bias to measure the more abundant peptides rather than the less abundant peptides only increases in multiplexed mass spectrometry. While this paper did not address multiplexed mass spectrometry, it is an increasingly important tool for interpreting protein phosphorylation in their physiological contexts and thus the compatibility of the reported smart polymer based enrichment method with this mass spectrometry method needs to be addressed.

Response: Thanks for this suggestion. For the HeLa S3 cell lysate, the selectivity of $\text{PNI-co-ATBA}_{0.2}\text{@SiO}_2$ towards phosphopeptides (PPs) was $82.3 \pm 4.1\%$, which means that about 17.7% identified peptides are non-phosphorylated ones (listed in Table S7 in the revised Supplementary Materials). These non-modified peptides (NMPs) were found in the Elution 2 solution, which was described in details in Supplementary Section 2.16. We have made clear description about the enrichment conditions. After checking the sequences of these NMPs, we found that the majority of NMPs contain multiple glutamic acids and/or aspartic acids located right next to each other, especially the top 20 abundant non-modified peptides co-eluted with

multi-phosphopeptides (Table S7).

Meanwhile, a new Supplementary Fig. S51 has been added, which shows the distribution of the signal intensities for the peptides with different numbers of phosphorylation sites and NMPs. It can be observed from Figure S51 that only a few NMPs are abundant compared with PPs. Among the PPs, the mono-PPs are the most abundant species, followed by di-PPs and tri-PPs. For case of PPs with more than 4 phosphorylation sites, they show the lowest signal intensities among all identified peptides but could also be detected efficiently. Corresponding description has been added to the manuscript in Page 16.

Regarding to the issue that some of the phosphopeptide has signal values of 0, we have re-processed the data and removed such data from the Table S6.

As to the issue of multiplexed mass spectrometry and phosphopeptide identification, we strongly agree with the referee that MS is a very important tool in proteomics. The introduction of the advanced multiplexed mass spectrometry will result in the identification of phosphorylation sites with much higher accuracy, but it will substantially reduce the number of phosphorylation sites.

In this work, we evaluated our material towards multiple phosphorylated peptide (MPP) enrichment with commonly employed HeLa cell lysate, reported database searching protocols, and mature MS data analysis strategy. The results have demonstrated that the prepared material has good potential for MPP enrichment. In the future, when the PNI-co-ATBA_{0.2}@SiO₂ is applied in large-scale phosphoproteomics studies, we will pay more attention to these issues including involvement of multiplexed mass spectrometry (as suggested by the Referee 1), and lower mass cut-off of product ions and rigid data searching parameters (as suggested by the Referee 2) to improve the accuracy of phosphorylation site identification.

Question 1b: On a related note, there is supplemental data in Figure S38 and Figure S39 that more clearly demonstrates the strengths of this method than what is currently shown in Figure 4. Specifically, panels C and D from Figure S38 could be shown along with panels E and F from Figure S39. The results from the 1:10 and 1:500

mixture experiments with PNI-co-ATBA @ SiO₂ currently in Figure 4 could be put instead in Figure S38. The authors could also choose a different color for labeling the 3P and 4P peptides in order to make the benefits of their method more clear.

Response: Thanks for this suggestion! We agree with your suggestion, however, the referee 3 presumed that the comparison of our polymeric material with commercially available TiO₂ was not equitable, because there are some improved TiO₂ or IMAC materials displaying certain MPP enrichment capacities. In order to eliminate the unnecessary misunderstanding for audiences, we delete the original panel 4e (TiO₂ enrichment performance) from the Figure 4 and add it to Supplementary Figure S45. According to your suggestion, the original panels 4b and 4e related to the separation of singly PP and multiple PP were also removed from Figure 4 and were added to Supplementary Figure S44. Accordingly, the MPP enrichment performance of our material from tryptic digests of casein mixed with 100- or 500-fold BSA were added to Figure 4, in order to better conform to the core topic of this paper.

Question 2: The first paragraph of the introduction (lines 26-38) is unacceptably weak and confusing. The authors do not make clear the distinction between multiple phosphorylations on a protein that would be seen on separate peptides and multiple phosphorylation sites where those phosphorylations would be found on the same peptide. The authors use retinoblastoma protein and Sic1 as examples where we know about the regulatory importance of multiple phosphorylations on a same peptide but the explanation of the “Rb phosphorylation codes” and the 6 sites of Sic1 that are phosphorylated are not explicit enough to make clear the importance of being able to measure single peptides that have multiple phosphorylations on them. The authors could also consider including a schematic of a protein with multiple phosphorylations very close to each other on sequence space is a first panel of Figure 1.

Response: Thanks for this suggestion! We have revised the corresponding description in the first paragraph and a new panel has been added to Figure 1, which clearly

displays the abundant phosphorylation sites located near the tubulin-binding domain of Tau protein.

Question 3: Clearly not all possible peptide sequences can be tested with model peptides, however, examination of the sequences of the peptides that the authors raises two concerns that should be mentioned in the text. Firstly, the sequences all include four prolines, raising the concern that their results might not apply to phosphorylations that are not proline directed. The authors should comment on the bias this sequence might represent. Secondly, the model acidic peptide NMP2 includes three aspartic acids but they are all separated in the peptide sequence. A quick scan of the peptide sequences reported in Table S6 shows that many of the phosphorylated peptides also include regions of poly (E) and poly (D). Readers who have done phosphoproteomics will know that many of the contaminating acidic peptide sequence include stretches of glutamic and aspartic acid residues right next to each other. The bias that their choice of acidic peptide introduces should be mentioned and it would be ideal to include a supplemental figure that shows data about the distribution of contaminating acidic peptide sequences that are seen in mass spectrometry during enrichment for phosphorylated peptides.

Response: Thanks for this suggestion! We agree with your comment. In the initial submission, we only studied two model non-modified peptides (NMPs) with identical peptide sequences to the model MPPs, but as you clearly pointed out and the data shown in Supplementary Table S7, there are some NMPs with high abundance co-eluted with MPPs, most of these NMPs contain multiple Asp and/or Glu residues right next to each other (also called poly(D) or poly(E)).

In order to evaluate the anti-interference capacity of our material towards these acidic NMPs, we introduce three new NMPs that are synthesized by China-Peptides Corp. (Shanghai) with high purity (> 99.5%), their peptide sequences are listed as below: NAESESEAEEDG, DLDAPDDVDF, YFQINQDEEEEEDED (NMP 3). The adsorption performance of these NMPs on our copolymer surface was studied by

QCM-D. As shown in Supplementary Fig. S17, these NMPs display evidential adsorption on the copolymer surface, and the adsorption-induced frequency changes (ΔF) are substantially larger than that of NMP 1 and 2. However, the maximal ΔF induced by NMP 3 is below 200 Hz, which is still lower than that induced by the MPP. Therefore, we presume that our material has satisfactory anti-interference capacity towards these acidic NMPs. However, in real biosamples, the abundance of these acidic NMPs is far substantially higher than that of MPPs, which will bring strong interference to the MPP identification. In the revised manuscript, NMP 3 is added as a typical example with multiple Glu residues and the corresponding description was added in Page 6, 7 and 16. Correspondingly, the adsorption curves of NMP 3 are added to Figure 2.

Question 4: The authors need to make clear their definition of the word “site” and the degree to which they are considering redundancy between peptides when quantifying the percentage of new sites and proportions of different peptides found in Figure 5. Inspection of Table S6 shows that the authors are calling each of the three phosphorylations on a tri-PP a different site, but this is not clearly stated in the paper. Additionally, inspection of the Table S4 shows that often missed cleavages of lysines on peptides or oxidations of methionine introduce modifications in the peptide sequence that are do not give more biological information about phosphorylations. (Peptides No.1 and No. 4 in Table S4 are an example of this.) If the authors are counting these peptides as two different di-peptides then they are misrepresenting the presence of biological meaningful diversity in the multi-PPs that they are recovering.

Response: Thanks for this suggestion. After considering your suggestions, we realized that phosphopeptides worksheet in original Table S6 are not really unique phosphopeptides. These phosphopeptides resulted directly from Maxquant only have peptide sequence but lacking the specific location of phosphorylation sites. Thus, we deleted this phosphopeptide worksheet and only keep the list of unique phosphosites worksheet, which contains both the sequence window and the phosphopeptides with

modified sites. Moreover, phosphopeptide with the same sequence and phosphosites but having other modifications (e.g., oxidation on methionine residue) was taken as one phosphopeptide. Peptides with the same amino acid sequence but different phosphorylation states were characterized as different phosphopeptides. We have re-analyzed the results from database searching again and revised the original Table S6 according. After pooling the results from three analytical repeats (requested by the referee 2), a total of 2525 unique phosphosites from 1257 unique phosphopeptides were characterized. We have revised this result in the manuscript accordingly.

Question 5: Minor Comments Figures Figure 1. Panel h. There should be an upward arrow before pH and T because some of the audience might not be as familiar with polymer physics as the authors. Line 399. The method used to measure the contact angle in panel E is not mentioned in the figure legend or the associated results section. Figure 2. Panels j, h. The plots should have the same Y-axis as the plots in a, b, d, e, g, and h. The plots should also have the same x-axis as the plots above or the reason why the experiments were done for a shorter amount of time should be stated. Figure 3. Panel a, b. Why is there no acetonitrile in the solution? Is that a typo or a constraint due to the AFM. If there is not acetonitrile in the solution it should be mentioned in the text since the importance of acetonitrile concentration is discussed extensively.

Response: Thanks for your suggestion!

In Figure 1 panel h, upward arrows have been added after pH and T.

The original Figure 1e related to water contact angle measurement has been removed to supplementary Fig. 15c, and the corresponding description of measurement method is added in SM Page S32.

For Figure 2, the adsorption curves of NMP3 have been added, and all the x-axis are adjusted to the same time scales.

For Figure 3a and 3b, we also investigated the morphological change of the copolymer film after treatment with a serine tetra-PP (4pS) solution in CH₃CN/H₂O

(v/v: 80:20) Tris-HCl buffer solution at pH 7.4, at 20 °C, for 1 h. The corresponding result is shown in Supplementary Fig. S32. Remarkable morphological change was also observed under this condition.

Question 6: Line 433. Is the use of the word “elution” a typo that should actually read “solution”. Elution does not make sense in the context of this figure. Figure 4. Panel a. The schematic for the enrichment and strategy should include the solution pHs and the temperature(s) at which the experiments are performed. Panels b, c, d. It appears that the Y-axis of Relative Abundance was determined by normalizing the signals by the largest signal associated with a phosphorylated peptide. If this is the case that should be stated, and if not the normalization method should be noted.

Response: Thanks for your suggestion! The word “elution” has been changed to “solution”.

The detailed pH values and temperature has been added to Figure 4a.

The description for the Y-axis of relative abundance has been added to each MS spectra.

Question 7: Figure 5. Panel c. It would be great if the Recovery experiment could be done with the NMP1 and NMP2. Panel d. The Y-axis should go to 100%. Panels c, d, f. Since the number of repeats for the experiment is not too large the data points themselves should be plotted (x-scatter can be added so that separate points resolved) rather than the bar-plot with the standard deviation bars. Panel e. The data should be represented with either a bar plot or table, but not with a pie chart. The fraction of non-phosphorylated peptides should also be included.

Response: Thanks for this suggestion! According to your request, we perform this experiment. Owing to weak adsorption of NMP 1 and NMP 2 on the materials, reliable recovery data could not be obtained through MS measurement. For NMP 3 with stronger binding with the materials, the recovery is measured to be 52 % and 28 %

for PNI-*co*-ATBA_{0.2}@SiO₂ or TiO₂, respectively. This data has been added to the revised Figure 5c, while the measured recovery for three replicates are shown independently. For the revised Figure 5d and 5f, the percentage of new phosphorylation sites and the percentage of Ser : Thr : Tyr is calculated from the result of total numbers of three analytical repeats, therefore, the standard deviation bars in the original figures have been removed. In addition, according to the referee's suggestion, the original Figure 5e (pie chart) has been changed to a table displaying both the identified numbers and percentages of 1PP—≥4PP.

Question 8: Text Line 64-65. Hydrogen bonds are often considered a type of electrostatic interaction to it is confusion to contrast H-bonding to both chelation and electrostatic interactions. Line 72. This sentence is confusing because it is not clear whether the molecules referred to in “biding or release of these molecules” are the same as the “guest molecules” mentioned earlier in the sentence. Line 75. The sentence on this line can be stronger.

Response: Corresponding description has been corrected.

Question 9: The paper does show that their ATBA copolymer is an efficient tool to modulate these behaviors and thus the “may” does not seem appropriate.

Response: Corresponding description has been corrected.

Referee Comments on the Supplemental Materials of Qing et al.

1. There are multiple times in the supplemental methods where the authors talk about washing a material a certain number of times and use the grammar “after washing for X times”. It should be “after washing X times”.

Reply: Thanks for this correction! We have revised these grammar mistakes.

2. 343-344 Can the authors give more information about “protease inhibitor cocktail” and “phosphatase inhibitor cocktail (P3)”? (P3)

Reply: Phosphatase inhibitor cocktail 3 (product number: P0044) and protease inhibitor cocktail (product number: P8340) were ordered from Sigma-Aldrich (St. Louis, MO, USA). The two kinds of inhibitors are supplied as a clear solution in dimethyl sulfoxide (DMSO). Phosphatase inhibitor cocktail 3 contains individual components with specific inhibitory properties. Cantharidin inhibits protein phosphatase, 2A-(-)-p-bromo-levamisole oxalate inhibits L-isoforms of alkaline phosphatases. Calyculin A inhibits protein phosphatases 1 and 2A. While protease inhibitor cocktail contains individual components including AEBSF, Aprotinin, Bestatin, E-64, Leupeptin and Pepstatin A. Each component has specific inhibitory properties. AEBSF and Aprotinin act to inhibit serine proteases, including trypsin, chymotrypsin, and plasmin amongst others. Bestatin inhibits aminopeptidases. E-64 acts against cysteine proteases. Leupeptin acts against both serine and cysteine proteases. Pepstatin A inhibits acid proteases.

We have added the corresponding information in the revised supplementary material (SM) Page S3.

3. Supplemental methods 2.15 - While the authors do give the concentrations of strong acids it would be good to also include measured pH of the solutions as well.

Reply: Thanks for this suggestion. We have added the measured pH of the solutions in the revised SM.

4. 361-362 - What were the rinse volumes increased to for the increased levels of BSA interference?

Reply: Thanks for this suggestion. The rinse volumes were 100 uL × 2 and 100 uL × 4 in order to remove non-modified peptides when levels of BSA interference were increased to 100 and 500 folds, respectively. We have this part in the revised SM.

5. 365-366 - The polymer to peptide mass ratio will be important to users of the method. Can the authors give more specifics about this besides that “the material was adjusted slightly to 2-3 mg” For the enrichment of PPs with commercially available TiO₂, what was the peptide starting material: enrichment material ratio?

Reply: Thanks for this suggestion. We did agree with the referee that peptide-to-beads ratio affect the enrichment selectivity of phosphopeptides. The amount of polymer material was 2.5 mg when 50 ug digested peptides from HeLa was loaded. The ratio of peptide starting amount to enrichment material is 1:50, while this ratio is 1:40 for commercially available TiO₂. The suggested peptide-to-beads of non-stimulated HeLa cell lysate is about 1:2 to 1:8 as stated in the literature (Li Q.R., Ning Z.B., Tang J.S., Nie S., Zeng R. *J. Proteome Res.*, **2009**, *8*, 5375–5381). We have revised the corresponding description accordingly.

6. Line 392 (and 398/399) the statement “The adsorption capacity was measured before the point that the PP signals were observed” is unclear. I would suggest saying, “the adsorption capacity was measured as the highest PP to adsorbing material ratio at which no PP signals were observed in the MS spectra”.

Reply: We have revised the corresponding description according to your suggestion.

7. 406 - The meaning of the word “targeted” is unclear.

Reply: This word has been changed to “the model peptide (1pS–4pS)”.

8. 483-484 It is unclear if the authors are talking about the work that the results of Figure S2 inspired them to do or if they are talking about how they are interpreting the results of Figure S2.

Reply: Thanks for this question! As a complement of ¹H NMR titration experiment, ¹³C NMR titration provides the information of chemical shifts of carbon atoms of ATBA in

complexation with HPO_4^{2-} . We have added some sentences to clarify this point.

9. Figure S5, S6 - What is $[\text{G}]/[\text{H}]$ stand for? What is $\text{Log}[\text{G}]$?

Reply: $[\text{G}]/[\text{H}]$ is an abbreviation of molar ratio of guest to host, $\text{log}[\text{G}]$ is the logarithmic value of guest anion concentration. The corresponding description has been added in the Supplementary Materials.

10. Figure S9. The pHs could be added to the inside right corner of each of the plots to make the figure easier to interpret.

Reply: In the Supplementary Fig. S10 and S11 in SM, the pH values have been added to the right corner of each of the plots.

11. Figure S14. it is unclear whether buffer was not used for both the experiment at different pHs or only the experiment where the peptides were also added. If there were no buffer for the measurements at different pHs, how was the pH maintained?

Reply: Thanks for this question! LCST of polymer is sensitive to environmental conditions, including the buffering agents. In this experiment, we adopt a typical LCST measurement method which been reported by numerous references. In order to avoid the potential impact of different buffering agents on LCST measurement, we only used pure water as the solvent, for pH 3 or 10, while appropriate amount of hydrochloric acid (pH 3) or sodium hydrate (pH 10) was added to adjust the solution pH value. In addition, the polymer solution was injected into a closed quartz cell and the LCST measurement could be completed within 1 hour, we confirm that the solution pH value would not change remarkably.

12. Figure S25. Should it be that different pH-mediated adsorption conversion windows were seen for the 3pY and 3pT? It is not clear why it is reported as two different windows for 3pY.

Reply: Thanks for this question! As you mentioned, different pH-mediated adsorption conversion windows were observed. The former one (indicated by yellow layer) could be attributed to the pH-mediated binding affinity change of ATBA with 3pY, strong acidity (lower than pH 3) would destroy the hydrogen bonding network of the copolymer and result

in the sharp decrease in 3pY adsorption quantity. The second adsorption conversion window (indicated by green layer) correspond to the conformation transition of the polymeric chain in response to the pH change, the adsorbed 3pY would be released from the contracted polymeric network. The former description in the Figure S25 caption is not clear, here, the above description has been added.

13. Figure S30. The differences in surface morphology should be quantified if you are going to draw conclusions from them.

Reply: Thanks for this question! In order to quantify the surface roughness, root mean square roughness—Rq value was introduced, the corresponding Rq values of the polymer surface have been added to the AFM images (Fig. S36 in SM).

14. Figure S43. The purpose of this figure is not clear and it is not clear what is meant by “new information”.

Reply: Thanks for this question! The original Figure S43 displays the distribution of molecular mass of PPs enriched with different methods, which is adopted from a classical references. Thus, no new information was provided. The former description of “new information provided by our material” brings a misunderstanding, we have deleted this subtitle.

15. Figure S45. This should be a table not a bar chart. There is no information added by showing the values as bars.

Reply: Thanks for this suggestion. We have revised Figure S45 to **Table S8** according to your suggestion

16. Figure S47. This data should be represented in a table

Reply: Thanks for this suggestion. We have revised original Figure S47 to Supplementary **Table S9** according to your suggestion.

Referee #2 (Remarks to the Author):

Guangyan Qing et al propose a new enrichment approach for multiply phosphorylated peptides based on smart polymer. Overall, this is a novel and interesting new approach with broad applications in the field of proteomics. The development and characterization of the material is well described. Probably that section of the manuscript could be shorter and instead the information provided in the supporting information section.

The key to this type of article is whether the technique performs well when dealing with real biological samples. The authors use a HeLa cell lysate to demonstrate the performance of their approach. This section of the manuscript is short. The following issues would need to be addressed:

Reply: Thanks for your positive comments! According to your suggestions, we have revised the manuscript accordingly, particularly a new Table S6 (Phosphorylation site information identified from HeLa S3 cell lysate) is uploaded for your checking, the detailed reversion and responding is listed as below:

Question 1. How many biological and analytical repeats were done? This seems to be a single analysis.

Response: Thanks for this comment. We have conducted two biological repeats, while for each biological repeat, three analytical repeats were performed. The phosphopeptide selectivity of two biological repeats was $79.4\% \pm 3.0\%$ (n=3) and $82.3 \pm 4.1\%$ (n=3), respectively. The average overlapping between two analytical repeats was 68%. Part of these results have been added to the revised manuscript.

In the initial submission, the number of phosphopeptides used in the manuscript was obtained from one single analysis. We really appreciate the referee's reminder. According to your reminder, we have pooled the data from the three analytical repeats and identified 2525 unique phosphorylation sites from 1257 unique PPs in 50 µg of HeLa S3 cell lysate (Supplementary Table S6). This part of result has been revised in the manuscript.

Question 2. When I analyzed table S6, it seems to me that phosphopeptides were reported more than once. When I compared the modified sequences, I only saw 695 unique phosphopeptides. The rest of the entries were repeats of the same sequences. Furthermore, 26 of the remaining 695 unique phosphopeptides had identical m/z and MS/MS IDs. This would mean that only 669 unique phosphopeptides are present in this table instead of the 1168 reported in the text. The authors need to address this issue.

Response: Thanks for this suggestion. We apology for making mistakes during data analysis. After considering your suggestions, we realized that phosphopeptides worksheet in original Table S6 were not really unique phosphopeptides. These phosphopeptides resulted directly from Maxquant only have peptide sequences, but lack the specific location of phosphosites. Thus, we delete this phosphopeptide worksheet and only keep the list of unique phosphorylation site worksheet, which contains both the sequence window and the phosphopeptides with modified sites. Moreover, phosphopeptide with the same sequence and phosphosites but having other modifications (e.g., oxidation on methionine residue) is taken as one phosphopeptide. Peptides with the same amino acid sequences but different phosphorylation states are characterized as different phosphopeptides. We have re-analyzed the results from database searching again according to your suggestion and revised the original Table S6.

Question 3: -of the remaining 669 unique phosphopeptides, 25 had charges of 4+ or more. Considering that the mass tolerance for the MS/MS was 0.8Da, I am not confident that these matches are right.

-of the remaining 669 unique peptides, 60 have mass errors on the parent ion > 1.5 ppm with some with over 5ppm. These seem to be outliers and the authors should verify that this is right.

Response: Thanks for this comment. We have processed the MS data again with the Maxquant software. Rigid parameters according to literatures are set. Only peptides

with 2, 3 and 4 charge states are selected for fragmentation, while the mass errors for precursor ions are set less than 5 ppm. The substantial revision has been made for the section S2.20 **MS data analysis** in Supplementary Materials.

Question 4: -The fact that the MS/MS were done in the trap could be problematic. In particular, considering that the tolerance on the MS/MS ions is 0.8 Da. It seems to me that the confidence would be much higher if the authors had performed MS and MS/MS at high resolution. This would be particularly the case for phosphopeptides with multiple phosphorylation sites.

Response: Thanks for this suggestion. We strongly agree with the referee's opinion that high confidence of phosphorylation sites can be obtained with MS and MS/MS at high resolution. For the case of performing MS and MS/MS at high resolution, higher energy collision dissociation (HCD) can realize this purpose. Our MS/MS data were acquired under collision-induced dissociation (CID). Concerning the fragmentation of peptides with CID and HCD, each collision mode has its advantages and disadvantages as reported by Mann's group and Gygi's group, respectively.^[1,2] Some advantages of HCD fragmentation over CID with Orbitrap include no low-mass cutoff and higher quality MS/MS spectra for confident phosphorylation site localization. A disadvantage of HCD is that spectral acquisition times are currently much longer than CID, resulting in decreased number of phosphopeptides. In the recently published large-scale phosphoproteomics published on *Science*, CID also been used.^[3] The aim of this study is to assess the efficiency of PNI-co-ATBA_{0.2}@SiO₂ toward multi-phosphopeptide enrichment. The results have demonstrated that the prepared materials have good selectivity for MPPs. In the future, when PNI-co-ATBA_{0.2}@SiO₂ is applied in large-scale phosphoproteomics study, we'll pay more attention to the issues provided by the referee 2 to improve the accuracy of phosphorylation site identification.

[1] Nagaraj, N.; D'Souza, R. C. J.; Cox, J.; Olsen, J. V.; Mann, M. Feasibility of large-scale phosphoproteomics with higher energy collisional dissociation

fragmentation. *J. Proteome Res.* **2010**, *9*, 6786–6794.

[2] Jedrychowski, M. P.; Huttlin, E. L.; Haas, W.; Sowa, M. E.; Rad, R.; Gygi, S. P., Evaluation of HCD- and CID-type fragmentation within their respective detection platforms for murine phosphoproteomics. *Mol. Cell. Proteomics* **2011**, *10*, M111.009910.

[3] Studer, R. A., *et al*, Evolution of protein phosphorylation across 18 fungal species. *Science* **2016**, *354*, 229–232.

Question 5: -The distribution of the Mascot score appears to be dependent on the number of phosphosites. Peptides with the highest number of phosphosites seem to have the lowest Mascot score. The author applied a Mascot score cutoff of 30. However, I am not sure that this is appropriate for phosphopeptides that have multiple phosphorylation sites.

Response: Thanks for this suggestion. We agree with the referee that the phosphopeptides with multiple sites should have lower score cutoff than the mono- and di-phosphopeptides. We sincerely apologize for the mistake when we wrote up the Data analysis section. The search engine used in this study was indeed Maxquant, but not Mascot, which could be confirmed from the original excel file (Table S6) and the raw data sent to the referee. We have revised this part accordingly. We think that the score 30 in Maxquant is appropriate for MPPs.

Question 6: -The method section for the database search and filtering need to be improved.

Response: Thanks for this suggestion. We have substantially revised this part accordingly.

Question 7: -Thank you for providing a few examples of the annotated MS/MS. They should all be provided for the unique phosphopeptides. It is likely that some of the MS/MS are from overlapping peptides. This would explain why some of the large ions in MS/MS are not annotated. The authors need to discuss this issue and its impact

on the results considering the 0.8Da mass tolerance in MS/MS.

Response: Thanks for constructive suggestion. After data searching with Maxquant again, the unique phosphopeptides are counted from the unique phosphosites worksheet. These unique phosphopeptides process both peptide sequences and the location of modified sites. We have deleted the phosphopeptide worksheet from the original Table S6 because they are not real unique phosphopeptides.

Concerning the 0.8 Da mass tolerance in MS/MS, we strongly agree with the referee that lower mass tolerance in MS/MS would reduce the number of overlapping peptides. From the perspective of material science that this paper mainly focuses on novel-material based methodology. The data from HeLa cell lysate evidently disclose the high selectivity of our material toward MPPS. But as pointed out by the referee, for large-scale phosphoproteomics and in-depth biological function studies, rigid parameters including lower mass tolerance for MS and MS/MS during data searching and processing will be the better choice. We will stick to these suggestions of referee 2 in the future application of this material.

Question 8: -The authors need to use other tools to validate their results and in particular the position of their sites. See Nat Biotechnol. 2013 Jun;31(6):557-64. doi: 10.1038/nbt.2585. Epub 2013 May 19.

Response: Thanks for this constructive comment. The purpose of this paper focuses on the design of novel smart polymer oriented to MPP enrichment. The percentage of MPPs is the key to the material assessment. We investigate the enrichment selectivity of PNI-co-ATBA_{0.2}@SiO₂ with the statistics methods. As suggested by the referee, for large-scale phosphoproteomics and in-depth biological function studies, it is necessary to use multiple tools to validate the identified phosphosites. We will pay particular attention on this issue in the following application of our material in large-scale phosphoproteomics analysis.

Referee 3:

To aim at specifically enriching multiple phosphorylated peptides (MPPS), authors synthesized a thermal-responsive polymer decorated with polar functional groups (ATBA) that are designed to interact with phosphorylated peptides through hydrogen binding. Although many techniques, such as IMAC, MOAC, strong cation exchange chromatography (SCX), strong anion exchange chromatography (SAX), and HILIC, have been well developed for the enrichment of phosphorylated peptides, authors state that their new method has two main advantages: (i) the method is based on hydrogen-bonding and has highly specific and efficient enrichment of MPPS, and (ii) the performance of their method is tunable.

Techniques that are based on specific affinity binding, e.g., MOAC and IMAC, are generally believed to have good specificity and efficiency for enriching phosphorylated peptides. Methods that rely on weak hydrogen bond, e.g., HILIC, are usually less specific and less efficient for enrichment of phosphorylated peptides. The results described in this study are contrary to the established understanding. Therefore, the authors must provide strong evidence and reasonable explanations to support their conclusions.

Overall, the experiments and results currently described in this paper are not sufficient to support the conclusions. When the advantages of the new method are clearly demonstrated and reasonably understood, a novel application of the method to studying a phosphorylation system of biological significance would strengthen the manuscript.

Reply: Thanks for these comments! MOAC and IMAC materials have been widely used in phosphoproteomics analysis and contribute most phosphorylation sites collected in various database. Different from these inorganic materials that are mainly based on chelation binding, hydrogen bond-based smart polymer displays some unique characteristics in MPP enrichment, which could become a useful supplementary to these inorganic enrichment materials.

On the other hand, separation of conventional HILIC materials strongly rely on the slight difference in the hydrophilicity among various analytes, while hydrophilic

interaction is usually regarded as a non-specific binding. We are familiar with this topic and have published tens of relevant papers on HILIC (e.g., Guo, Z. M., Liang, X. M., *et al.* “Click saccharides”: novel separation materials for hydrophilic interaction liquid chromatography. *Chem. Commun.* **24**, 2491-2493 (2007); Liang, X. M. *et al.* A novel zwitterionic HILIC stationary phase based on “thio-ene” click chemistry between cysteine and vinyl silica. *Chem. Commun.* **47**, 4550-4552 (2011)). As for this study, our polymeric material cannot be assigned to hydrophilic material because the polymeric surface is hydrophobic with a water contact angle of 80°, which is beyond the definition of hydrophilic surface.

Importantly, as you known, although an individual hydrogen bond is weak and lacks selectivity, a specially designed molecule system with multiple hydrogen bonds and special configuration could display excellent selectivity towards a guest molecule, which has been well acknowledged in host-guest chemistry. Based on this concept, ATBA molecule is designed to bind with phosphate group with high affinity and selectivity, and the combination of ATBA with PNIPAAm further strengthen this capacity, leading to good performance in MPP enrichment.

Exactly as pointed out by the referee, preparation of phosphate guest, molecular basis for phosphate recognition, binding affinity between polymer and peptides, impact of ATBA grafting density on MPP adsorption, and other experimental details were not described clearly or not fully discussed in the initial submission. In the revised manuscript, according to the referee’s questions, a series of supplementary experiments are performed and the corresponding corrections or additional descriptions have been added. Again, thanks for the referee’s helpful comments!

Question 1. Modulation of binding of phosphorylated peptide by using solution conditions, such as solvent polarity and pH, is not unique to this method. Such modulation by changing solution conditions can also be achieved using other enrichment methods.

Reply: Thanks for this comment! In conventional chromatographic separation and PP

enrichment methods, solvent polarity and solution pH are widely applied to mediate the retention behaviors of various analytes, thus we agree with your comment. However, in addition to such knowledge, these traditional chromatographic parameters could be used to mediate the overturn of polymer chain, the *coil-to-globule* transition of the polymer conformation, which brings obvious advantages in high controllability of MPP adsorption/desorption. For example, as described in manuscript Page 7, “This narrow adsorption conversion window is fundamentally different from the typical hydrophilic interaction modes in liquid chromatography, which usually shows an approximately linear relationship between solvent polarity and chromatographic retention capacity of analyte ^[39].”

We believe that the combination of solvent, pH-mediated polymer chain transition and controllable MPP adsorption/desorption behaviours is seldom reported before, which provides a new insight for the development of MPP enrichment materials.

Question 2. Authors reasoned that high specificity and efficiency of the method for enriching MPPS result from involvement of multiple complementary H-bonds. However, authors did not provide K_a values of interactions between the polymer and different model peptides. The K_a values provided on the interactions between monomer ATBA and different model peptides cannot represent interactions involving multiple phosphate groups of a single peptide and multiple ATBA groups of a single polymer.

Reply: Thanks for this suggestion! According to your request, we calculate the association rate constants (K_a) of peptides adsorbed on PNI-*co*-ATBA_{0.2} surface by surface plasma resonance (SPR) adsorption experiment, which is a real-time and label free tool for investigating molecule-molecule interactions. The K_{ass} of 4pS, 1pS and NMP 2 adsorbed on copolymer surfaces are 1280, 294 and 145 $M^{-1}s^{-1}$, respectively, which reveal the differential binding affinity of the copolymer towards these peptides. A new Figure S13 and corresponding experiment details have been added to supplementary materials, meanwhile, a short description has been added to the

manuscript in Page 5.

Question 3. Polymers having different contents of ATBA should be used to study their binding with different peptides, which would reveal how the density of ATBA groups affects the interaction.

Reply: Thanks for this suggestion! Grafting density of ATBA in copolymer is a critical parameter for controlling the conformation of the polymer chain. Before determining the optimal polymer composition, we prepared a series of copolymers (PNI-*co*-ATBA_{0.10, 0.15, 0.2, 0.25, 0.35}) and investigated their MPP adsorption capacities. As shown in Figure S27 in SM, 0.20 was determined to be the optimal grafting density of ATBA owing to the highest 4pS adsorption capacity. To explain the possible adsorption mechanism, a new Figure S28 has been added to SM. Low ATBA grafting densities (i.e., 0.10, 0.15) would lead to insufficient PP binding sites, by contrast, high ATBA grafting densities (i.e., 0.25, 0.35) lead to a highly compact polymer network, only a small amount of ATBA unit expose outside. Accordingly, a short paragraph has been added to the revised manuscript (Page 10) to explain this effect, which could help audiences better understand the design of smart polymer.

Question 4. Because ATBA is randomly distributed in the linear polymer and phosphate groups are also present at different locations in the peptides, how specific can this method achieve the binding of different peptides containing multiple phosphate groups at different amino acid sites?

Reply: Thanks for this comment! As you mentioned, PNI-*co*-ATBA_{0.2} is a random copolymer that lacks definite secondary conformation like that of helical polymers or polymeric aggregates. Owing to the inherit characteristic of random copolymer, we could not provide a definite binding mode to describe how the polymer bind with multiple phosphate groups in MPPs. However, according to the experimental data, we presume that the high specificity of material towards MPPs originate from the core

ATBA recognition unit, while the flexible polymer chains further provide high-density of ATBA capable of binding with MPPs, which substantially improve the MPP binding affinity of the polymer. In order to clarify this point, a new Scheme S5 has been added to SM.

As for the selective binding of copolymer with Ser, Thr, or Tyr-modified PP, such selectivity could be attributed to the different binding capacity of ATBA with hydrogen phosphate ($K_a: 5.34 \times 10^4 \text{ L} \cdot \text{mol}^{-1}$) and benzyl phosphate ($K_a: 7.80 \times 10^4 \text{ L} \cdot \text{mol}^{-1}$), which was further proven by the binding affinity between ATBA and 3pS ($K_a: 1.50 \times 10^4 \text{ L} \cdot \text{mol}^{-1}$) and 3pY ($K_a: 1.82 \times 10^4 \text{ L} \cdot \text{mol}^{-1}$), as shown in Figure S12 in SM. Meanwhile, the K_a of ATBA with 3pS was larger than that with 3pT, we presume that this could be attributed to smaller steric hindrance of Ser than that of Thr.

Question 5. Page 4, line 88-93, the K_a of ATBA monomer to hydrogen phosphate ($K_a: 5.34 \times 10(4) \text{ L/mol}$) was only seven times higher than that with acetate (main source of acidic peptide, $K_a: 7.6 \times 10(3) \text{ L/mol}$). Considering that two amino acids contain acetate groups in their side chain, the concentration of peptides containing multiple acetate groups could be significantly higher than that of MPP. Therefore, how does such small difference in binding affinity allow for differentiating MPP from peptides containing multiple acetate groups? Based on such difference in K_a value, how will authors explain their results that the polymer is able to enrich PPs from 1:500 ratio of BSA?

Reply: This is a good question! In the initial submission, the association constants of ATBA with various anions were evaluated in two kinds of solvents, one is Tris-buffer aqueous solution, the other is a pure organic polar solvent—DMSO. In Tris-buffer solution, the K_a ratio was approximately 7:1 for HPO_4^{2-} and CH_3COO^- (Table S1 in SM); however, this K_a ratio remarkably increased to 1150:1 when the binding affinity was evaluated in DMSO (Table S3 in SM). These data indicated that ATBA had better discrimination capacity in organic solvent than that in water, which is in accordance with many references.

Considering the real PP adsorption and enrichment condition, the binding affinity of ATBA towards various anions was also evaluated in CH₃CN/H₂O (v/v: 80:20) Tris-HCl buffer solution (10 mM, pH: 7.4), as shown in Figure S6 and Table S2 in SM. Under this condition, the K_a ratio between HPO₄²⁻ and CH₃COO⁻ is 45:1 for ATBA, such binding difference could be amplified remarkably when ATBA molecules are integrated into a polymer system, which provides a solid foundation for the excellent recognition capacity of our material towards MPPs, even under the high ratios of BSA interferences.

Question 6. Results in Figure 4e and Figure 5c show very poor or no interaction of 4pS with TiO₂. However, the enrichment of MPP using TiO₂-based materials have been well demonstrated by many groups. (Wan, J.J., et al., TiO₂-Modified Macroporous Silica Foams for Advanced Enrichment of Multi-Phosphorylated Peptides. *Chemistry-a European Journal*, 2009. 15(11): p. 2504-2508. Yue, X.S., A. Schunter, and A.B. Hummon, Comparing Multistep Immobilized Metal Affinity Chromatography and Multistep TiO₂ Methods for Phosphopeptide Enrichment. *Analytical Chemistry*, 2015. 87(17): p. 8837-8844. Zhao, X.Y., et al., Citric Acid-Assisted Two-Step Enrichment with TiO₂ Enhances the Separation of Multi- and Monophosphorylated Peptides and Increases Phosphoprotein Profiling. *Journal of Proteome Research*, 2013. 12(6): p. 2467-2476.)

Reply: Thanks for this comment! As you mentioned, some references reported the improved TiO₂- or IMAC materials, which displayed satisfactory enrichment capacities towards MPPs. In this work, commercially available TiO₂ (GE Corp.) is introduced to compare with our material, maybe it is not reasonable because we could not obtain these improved TiO₂ materials and evaluate their PP performance one by one. In order to eliminate this misunderstanding, the original Figure 4e has been removed from the manuscript and added to Figure S45 in SM, as a reference for audiences. Meanwhile, four relevant references have been added to the manuscript to introduce these improved IMOC or IMAC materials for MPP enrichment.

For Figure 5c, TiO₂ has satisfactory recovery towards 1pS and 2pS, by comparison, our copolymer material has better performance towards 3pS and 4pS, which reveal the good complementarity between TiO₂ and our materials.

Question 7. Page 6, line 134, TiO₂ or ZrO₂ have much weaker adsorption of PPs. Why is the recovery of 1-3Ps from the use of TiO₂ and polymer comparable (Figure 5C)?

9. In Figure 5, why is the recovery of 4pS so low on TiO₂? Does this mean that TiO₂ cannot capture 4pS, but can capture 1pS, 2pS and 3pS? However, the adsorption capacity of TiO₂ for 4pS is higher than for 1pS, 2pS and 3pS (Figure 5a). How would authors explain this inconsistency?

Reply: Thanks for these questions! As you know, PP adsorption capacity and recovery are two concepts in phosphoproteomics analysis. PP adsorption capacity is defined as the maximum adsorption quantity of a PP on a material, which only corresponds to the PP binding process. While PP recovery is defined as the ratio of released PP to the total PP, involving in both PP binding and releasing processes. Benefited from chelation bonds with PPs, TiO₂ or ZrO₂ has strong binding with PPs and this affinity increases with the degree of phosphorylation owing to more binding sites, thus TiO₂ has stronger binding with 4pS than that of 1pS—3pS.

However, this chelation binding mode also brings an obvious drawback that only a small amount of bound MPP (i.e., 3pS or 4pS) could be dissociated from the TiO₂ surface, resulting in low recoveries for MPPs. By comparison, owing to strong adsorption capacity towards MPPs and highly controllable binding behaviors of our material, most of bound MPPs could be disassociated from our copolymer surface, leading to high recoveries for MPPs. Therefore, For 1pS and 2pS, the recoveries of our material are comparable to TiO₂, but are much higher than TiO₂ when 3pS and 4pS are evaluated. In the revised manuscript Page 15, we have added the above explanation.

Question 8. As shown, in the fluorescence titration experiment (Figure S5), up to 150 μM of H_3PO_4 , H_2PO_4^- , HPO_4^{2-} and PO_4^{3-} were added into 10 mM TrisHCl (pH 7.4). With the Tris buffer at pH 7.4, the composition of phosphate ions should be similar. How would authors explain the differences in K_a ?

Reply: Thanks for this question! Anion recognition is a hot research topic in host-guest chemistry, we are familiar with this topic. In order to eliminate the impact of cation and its stoichiometry on K_a measurement, H_3PO_4 was allowed to react with different molar ratios of tetrabutylammonium hydroxide to prepare various phosphate anions, and then these anions (the bulky tetrabutylammonium work as cations) were added to Tris-buffer solutions, respectively.

Considering the respective ionization equilibrium of each phosphate anion, for a specific anion, slight difference in anion concentration indeed exists between the calculated value and the actual one. However, the composition difference among H_3PO_4 , H_2PO_4^- , HPO_4^{2-} and PO_4^{3-} is remarkable. This anion preparation method has been used by most relevant host-guest chemistry references (e.g. Martinez-Manez, R. & Sancenon, F. Fluorogenic and chromogenic chemosensors and reagents for anions. *Chem. Rev.* **103**, 4419-4476 (2003)), which provides an ideal environment to investigate the interaction between anion and its receptor. In the initial submission, we neglected to describe the preparation method of anion guests, in the revised supplementary materials, we have supplied this information in SM Page S8.

Question 9. A significant application should be demonstrated, such as analysis of samples with clinical/biological significance.

Reply: Thanks for this suggestion! In this work, we attempt to report a new design idea for MPP enrichment, the preliminary experiment data could prove the application potential of our material. We clearly realize that this work is just a beginning, a more challenging work is how to use our material tool to discover valuable MPP sites and build a clear relationship between the sites and their biological functions. Right now

we are cooperating with the biologists and doctors in Tongji Medical School of Huazhong University of Science and Technology), relevant research progress will be reported in near future. We think this biological research could become an independent work with greater significance. Overall, thanks for your positive and helpful comments, which help us improve our work to satisfy the high demand of this journal.

REVIEWERS' COMMENTS:

Reviewer #1 (Remarks to the Author):

All of the technical requests were addressed appropriately. However, I think that the response to Question 2 in the paper needs to be improved. The authors did make changes to the first introductory paragraph and added a new panel to Figure 1, but the sentence starting with "Benefited from multisite phosphorylation, ..." is confusing and unreadable. It seems that the goal of the sentence is to say that multisite phosphorylation is an important mechanism for regulating protein activity. It would be just as effective to say something like (following the discussion on Tau protein) -- "In addition to pathological significance, there are many examples where adjacent multisite phosphorylations regulate protein activity (citations)". This section **MUST** be improved to a readable and clear state - this first stab is a small move in the right direction but more work is needed here.

Also, on line 412 another sentence begins with "Benefited from" and it should read "Benefiting from ..."

Reviewer #2 (Remarks to the Author):

I am fine with the revisions from the authors.

Reviewer #3 (Remarks to the Author):

The authors have addressed my concerns. The revised manuscript included additional supporting data and provided necessary clarifications. The quality of the revised manuscript is much improved. I support its publication.

Reply to the Reviewers' questions:

Reviewer #1 (Remarks to the Author):

All of the technical requests were addressed appropriately. However, I think that the response to Question 2 in the paper needs to be improved. The authors did make changes to the first introductory paragraph and added a new panel to Figure 1, but the sentence starting with "Benefited from multisite phosphorylation, ..." is confusing and unreadable. It seems that the goal of the sentence is to say that multisite phosphorylation is an important mechanism for regulating protein activity. It would be just as effective to say something like (following the discussion on Tau protein) -- "In addition to pathological significance, there are many examples where adjacent multisite phosphorylations regulate protein activity (citations)". This section MUST be improved to a readable and clear state - this first stab is a small move in the right direction but more work is needed here.

Also, on line 412 another sentence begins with "Benefited from" and it should read "Benefiting from ..."

Reply: We appreciate the great help from the Reviewer 1! The former description for the significance of adjacent multisite phosphorylation was still not specific and might bring some misunderstanding. After discussing with two experts in Phosphoproteome and molecular biology, we add two specific examples, including a recent work published on *Science* (Ref. 11) to the introduction section, which could better display the essential roles of adjacent multisite phosphorylation of peptides.

Other grammar problems have been addressed.

Reviewer #2 (Remarks to the Author):

I am fine with the revisions from the authors.

Thanks for your professional and helpful comments!

Reviewer #3 (Remarks to the Author):

The authors have addressed my concerns. The revised manuscript included additional supporting data and provided necessary clarifications. The quality of the revised manuscript is much improved. I support its publication.

Thanks for your professional and helpful comments!